# *A Stitch in Time Forecasts Nine*: Towards End-to-End Agentic Time Series Forecasting

## Abstract

Time series forecasting is central to decision-making in domains as diverse as energy, finance, climate, and public health. In practice, forecasters face thousands of short, noisy series that vary in frequency, quality, and horizon, where the dominant cost lies not in model fitting, but in the labor-intensive preprocessing, validation, and ensembling required to obtain reliable predictions. Prevailing statistical and deep learning models are tailored to specific datasets or domains and generalize poorly. A general, domain-agnostic framework that minimizes human intervention is urgently in demand. In this paper, we introduce **TimeSeriesScientist** (**TSci**), the first LLM-driven agentic framework for general time series forecasting. The framework comprises four specialized agents: *Curator* performs LLM-guided diagnostics augmented by external tools that reason over data statistics to choose targeted preprocessing; *Planner* narrows the hypothesis space of model choice by leveraging multi-modal diagnostics and self-planning over the input; *Forecaster* performs model fitting and validation and based on the results to adaptively select the best model configuration as well as ensemble strategy to make final predictions; and *Reporter* synthesizes the whole process into a comprehensive, transparent report. With transparent natural-language rationales and comprehensive reports, TSci transforms the forecasting workflow into a white-box system that is both interpretable and extensible across tasks. Empirical results on eight established benchmarks demonstrate that TSci consistently outperforms both statistical and LLM-based baselines, reducing forecast error by an average of **10.4%** and **38.2%**, respectively. Moreover, TSci produces a clear and rigorous report that makes the forecasting workflow more transparent and interpretable. Our codes are available at Anonymous GitHub for reproducibility.

## 1 Introduction

Time series forecasting guides decision making in domains as diverse as energy (Liu et al., 2023), finance (Zhu & Shasha, 2002), climate (Schneider & Dickinson, 1974), and public health (Matsubara et al., 2014). In practice, organizations manage tens of thousands of short, noisy time series data with heterogeneous sampling, missing values, and shifting horizons (Makridakis et al., 2020; Taylor & Letham, 2018; Makridakis et al., 2022). The dominant cost in forecasting is often not model fitting, but rather building reliable data processing and evaluation pipelines. This process is non-trivial for short and noisy series with irregular sampling and intermittent observations, and they remain largely manual in practice (Tawakuli et al., 2025; Shukla & Marlin, 2021; Moritz & Bartz-Beielstein, 2017). Despite the availability of strong libraries that streamline modeling itself (Alexandrov et al., 2019; Herzen et al., 2022; Jiang et al., 2022), end-to-end pipelines still require substantial human effort to tailor preprocessing, validation, and ensembling to each new collection of series.

Most advances in forecasting now arrive as expert models tuned to specific domains, or universal approaches that optimize only the model while leaving the rest of the pipeline untouched (Shchur et al., 2023; Gruver et al., 2024; Roque et al., 2024). Such systems can reach SOTA in-domain performance yet degrade under distribution shift because they rely on dataset or distribution-specific tuning rather than generalizable reasoning about the series (Zhang et al., 2023a). AutoML for forecasting (Shchur et al., 2023) centers on model selection and ensembling, but with limited attention to data quality. And it lacks the capacity to *reason* about temporal structure, adapt tools to heterogeneous series, and justify choices in natural language. Meanwhile, Time-LLM (Jin et al., 2023)

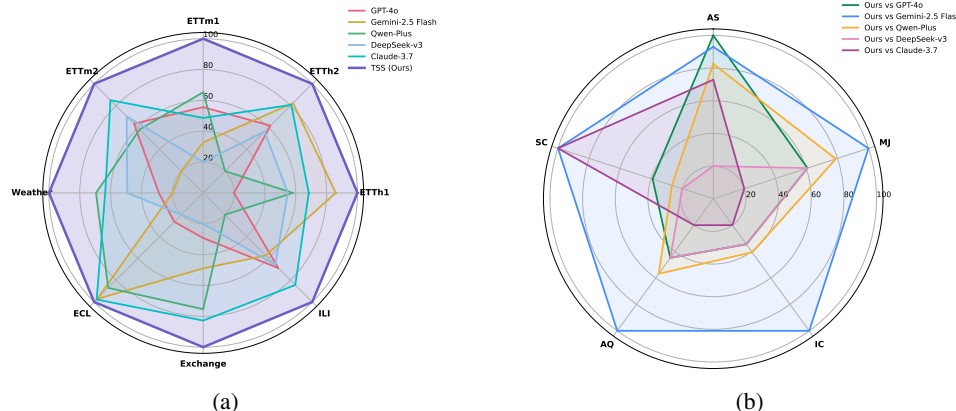

Figure 1: **Performance comparison of TSci with five LLM-based baselines**. TSci outperforms LLM-based baselines on eight benchmarks spanning five domains (Figure 1a). The comprehensive report generated by TSci outperforms LLM-based baselines across five rubrics (Figure 1b).

achieves strong in-domain performance, yet it still primarily targets the model rather than the end-to-end pipeline (Gruver et al., 2024). These limitations motivate an *agentic* approach, one that treats time series forecasting as a sequential decision process over data preparation, model selection, validation, and ensembling, with explicit planning, tool use, and transparent rationales.

To this end, we introduce **TimeSeriesScientist (TSci)**, the first end-to-end, agentic framework that leverages multimodal knowledge to automate the entire workflow a human scientist would follow for univariate time series forecasting. Rather than committing to a single universal model, TSci orchestrates four specialized agents throughout the process. First, *Curator* performs LLM-guided diagnostics augmented by external tools that reason over data statistics. It generates a compact set of visualizations leveraging LLM multimodal ability and outputs an analysis summary of temporal structure that guides subsequent steps. Next, *Planner* selects candidate models from a predefined model library based on the multimodal diagnostics and optimizes hyperparameters through a validation-driven search. Then, *Forecaster* reasons over validation results and adaptively selects an ensemble strategy to produce the final prediction along with natural-language rationales. Finally, *Reporter* consolidates all intermediate statistical analyses and forecasting results and outputs a comprehensive report. This design transforms forecasting into an adaptive, interpretable, and extensible pipeline, bridging the gap between human expertise and automated decision-making.

Across eight public benchmarks spanning five domains, TSci consistently outperforms both statistical and LLM-driven baselines, reducing forecasting error by **10.4%** and **38.3%** on average, respectively. Ablations show that each module contributes materially to the performance. Our evaluation of the report generator further demonstrates its technical rigor and clear communication, supporting practical deployment in settings that demand transparency and auditability.

**Our main contributions are as follows**: 1) We introduce **TimeSeriesScientist**, the first end-to-end, agentic framework for univariate time series forecasting with tool-augmented LLM reasoning; 2) We propose plot-informed multimodal diagnostics, where a lightweight vision encoder converts plots into descriptors guiding preprocessing, analysis, and model selection; 3) We show that TSci outperforms both statistical and LLM-diven baselines across diverse benchmarks; and 4) We provide a comprehensive evaluation of its generated reports, demonstrating both technical rigor and communication quality.

## 2 RELATED WORK

**Time Series Forecasting.** Univariate time series forecasting has evolved from classical statistical methods (e.g., ARIMA, ETS, and TBATS), which exploit linear trends and seasonalities (Box et al., 2015; Hyndman & Khandakar, 2008; De Livera et al., 2011), to global deep learning models (e.g., DeepAR, N-BEATS, and PatchTST) that capture nonlinear patterns and long-term dependencies (Salinas et al., 2020; Oreshkin et al., 2019; Nie et al., 2023). More recently, foundation-style approaches (e.g., Chronos, TimesFM, Lag-Llama) and prompt-based adaptations of LLMs (e.g.,

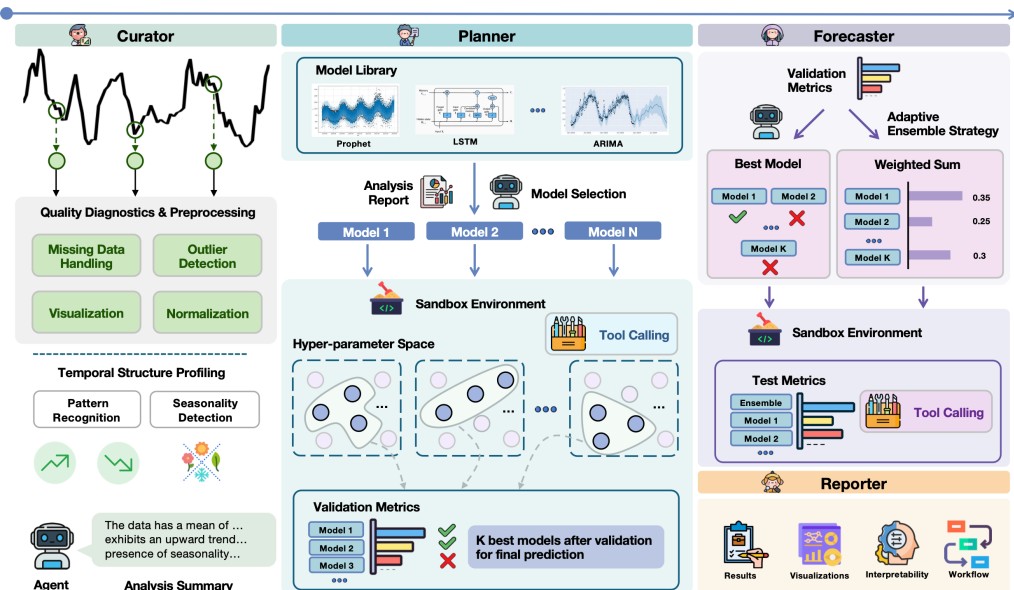

Figure 2: **Overview of our proposed TSci framework.** This collaborative multi-agent system is designed to analyze and forecast general time series data, just like a human scientist. Upon receiving input time series data, the framework executes a structured four-agent workflow. Curator generates analytical reports (Section 3.2), Planner selects model configurations through reasoning and validation (Section 3.3), Forecaster integrates model results to produce the final forecast (Section 3.4), Reporter generates a comprehensive report as the final output of our framework (Section 3.5).

GPT4TS, Time-LLM) have demonstrated zero-shot and few-shot forecasting capabilities (Ansari et al., 2024; Das et al., 2024; Rasul et al., 2023; Zhou et al., 2023; Jin et al., 2023), treating time series as sequences to be modeled in analogy with language. While these advances highlight a trend toward general-purpose and transferable forecasters, existing work remains largely model-centric: the broader pipeline of preprocessing, evaluation design, and ensemble synthesis continues to rely heavily on manual effort. This gap motivates our pursuit of an end-to-end, LLM-powered agentic framework that integrates reasoning, tool use, and automation across the entire forecasting workflow.

**Multi-agent System.** Large language models have enabled the rise of multi-agent systems, where specialized agents collaborate via communication and tool use to tackle complex analytical tasks. Frameworks such as CAMEL (Li et al., 2023), AutoGen (Wu et al., 2023b), and DSPy (Khattab et al., 2024) demonstrate how planner–executor architectures can coordinate agents for reasoning, retrieval, and problem solving (Khattab et al., 2024). Recent applications show their utility for domains like business intelligence and financial forecasting (Wawer & Chudziak, 2025). Despite this progress, existing systems rarely address the unique challenges of time series: heterogeneous sampling and multimodal data that are often irregular or asynchronous (Chang et al., 2025), and the need for transparent ensemble reporting of forecasts (Zhao & jiekai ma, 2025). This leaves open the opportunity for a multi-agent, domain-agnostic framework that leverages LLM reasoning to automate forecasting pipelines while ensuring interpretability and auditability.

## 3 TIMESERIESSCIENTIST

TSci acts as a human scientist, having the ability to systematically perform data analysis, model selection, forecasting, and report generation by utilizing LLM reasoning abilities. TSci integrates four specialized agents, each assigned a distinct role, and collaboratively engages in the whole process: (1) *Curator*: Performs LLM-guided diagnoses augmented by external tools that reason over data statistics and output a multimodel summary guiding subsequent steps; (2) *Planner*: Narrows the model configuration space by leveraging multimodal diagnostics and a validation-driven search; (3) *Forecaster*: Reasons over validation results to adaptively select model ensemble strategy and produces the final forecast; and (4) *Reporter*: Generates a comprehensive report consolidating all intermediate statistical analyses and forecasting results.

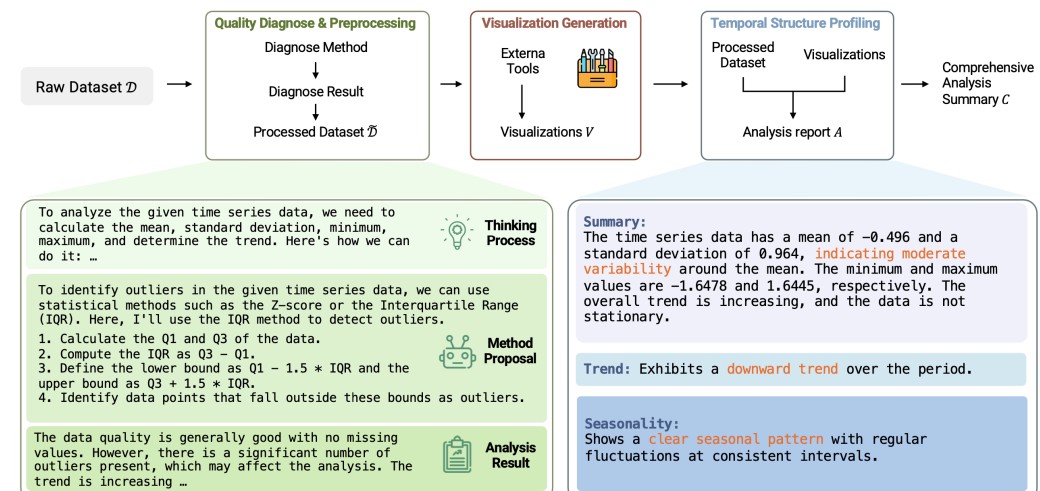

Figure 3: **Workflow of Curator.** The raw dataset $\mathcal{D}$ is first diagnosed and processed into a cleaned dataset $\tilde{\mathcal{D}}$. Next, the agent generates tailored visualizations $V$ to expose temporal structures and facilitate interpretability. Finally, the agent integrates the processed data and visualizations to extract trends, seasonality, and stationarity, producing a comprehensive analysis summary $S$.

### 3.1 PROBLEM FORMULATION

We first formally formulate the univariate time series forecasting problem. Let $\mathbf{x} = \{x_{t-T+1}, ..., x_{t-1}, x_t\} \in \mathbb{R}^{1 \times T}$ be a given univariate time series with $T$ values in the historical data, where each $x_{t-i}$, for $i = 0, ..., T-1$, represents a recorded value of the variable $\mathbf{x}$ at time $t - i$. The forecasting process consists of estimating the value of $y_{t+i} \in \mathbb{R}^{1 \times H}$, denoted as $\hat{y}_{t+i}$, $i = 1, ..., H$, where $H$ is the horizon of prediction. The overall objective is to minimize the mean average errors (MAE) between the ground truths and predictions, i.e., $\frac{1}{H} \sum_{i=1}^{H} ||y_{t+i} - \hat{y}_{t+i}||$.

In our proposed framework, given a univariate time series data $\mathcal{D}$, the system generates a comprehensive report $\mathcal{R}$ containing: statistics of the input data, visualizations, proposed model combinations that best fit the data, and the final forecasting result. This framework significantly reduces manual effort and time cost, while providing human scientists with a detailed and reliable analytical output.

### 3.2 CURATOR

Data preprocessing is critical in time series forecasting, as it ensures data quality, improves model accuracy, and directly impacts the reliability of analytical results (Chakraborty & Joseph, 2017; Esmael et al., 2012; Zhang et al., 2022; Shih et al., 2023). Curator leverages LLM reasoning ability, augmented with specialized tools to transform the raw series into a clean and informative form that downstream agents can depend on. It operates in three coordinated steps. Details are in Figure 3.

**Quality Diagnostics & Preprocessing.** High-quality input is critical for reliable forecasting. Rather than computing fixed summaries, Curator leverages LLM-driven reasoning to both *diagnose* issues and *execute* appropriate preprocessing. Specifically, given a univariate series $\mathcal{D} = \{x_t\}_{t=1}^{T}$, the agent first outputs a vector $Q$ containing data statistics $S$, missing value information $M$, outlier information $O$, and data-process strategy $\pi$. This process can be formalized as:

$$Q = \mathcal{A}_f(\mathcal{D}) = \left( S, M, O, \pi \right), \tag{1}$$

where $\mathcal{A}_f$ denotes the quality diagnostics operator, $S = (\mu, \sigma, x_{\min}, x_{\max}, \tau_{\text{trend}})$ denotes basic data statistics containing mean, standard deviation, min/max value, and trend, $\pi = (m^*, h^*)$ denotes LLM-recommended missing value and outlier handling strategies.

Based on processing strategy $\pi$, the agent applies transformation $\phi \colon \mathbb{R}^T \to \mathbb{R}^T$ to the raw input series $\mathcal{D}$, and get a processed series $\tilde{\mathcal{D}} = \phi(\mathcal{D}) = \{\tilde{x}_t\}_{t=1}^{T}$, where $\tilde{x}_t$ denotes the processed value at time step $t$. By coupling quality diagnostics with preprocessing, the agent tailors data-aware

strategies, yielding a well-conditioned preprocessed dataset that supports subsequent steps. Details about strategies and transformations can be found in Appendix A.

**Visualization Generation.** Visualizations greatly aid human scientists in comprehending complex time series data and identifying critical temporal patterns. Inspired by this practice, the agent automates the creation of insightful visualizations leveraging natural language prompts and reasoning from an LLM. This step can be formalized as generating a visualization suite given a processed dataset: $V = \mathcal{A}_v(\tilde{\mathcal{D}})$, where $\mathcal{A}_v$ denotes the visualization generator. Specifically, it generates three primary visualization types tailored to input data characteristics: (1) Time series overview plot: Visualize data statistics, illustrate moving averages and standard deviations. (2) Time series decomposition analysis plot: Reveals temporal patterns, long-term trends, and seasonal cycles. (3) Autocorrelation analysis plot: Identify temporal dependencies, detect non-stationarity, and guide the later selection of appropriate model parameters. Details about the plots are provided in Appendix E.

**Temporal Structure Profiling.** To effectively support downstream forecasting, an overall analysis is important in uncovering temporal structures and statistical properties that are essential for informed model selection and interpretation. This step conducts analysis through prompting to extract meaningful patterns and features from preprocessed time series data. The objective is to detect trends, seasonality, and stationarity, thereby guiding the selection of suitable forecasting models. Formally, given the processed dataset $\tilde{\mathcal{D}}$ and visualizations $V$, the agent generates an analysis report $A$ through LLM reasoning: $A = \mathcal{A}_c(\tilde{\mathcal{D}}, V) = \{t, s, u\}$, where $\mathcal{A}_c$ denotes the profiling step, $t, s, u$ denote trend, seasonality, and stationary, respectively.

The outcome of Curator is a comprehensive analysis summary $C = \{Q, V, A\}$, where $Q, V, A$ are the outputs from each step, respectively.

## 3.3 PLANNER

Planner narrows the hypothesis space of model configurations by reasoning on the analysis summary $C$. Rather than exhaustively trying all candidates, it prioritizes models that are most consistent with data characteristics. Concretely, Planner operates in three coordinated steps.

**Model Selection.** Planner extracts visual features from visualizations $V$ via lightwise pattern recognition and LLM reasoning. It then maps the recognized data pattern to suitable model families and forms a candidate pool $\mathcal{M}_p$, which has $n_p$ candidate models from a pre-defined model library $\mathcal{M}$: $\mathcal{M}_p = \text{Select}(\mathcal{M}; n_p)$, where $|\mathcal{M}_p| = n_p$. Concretely, the agent may choose to use Prophet when recognizing a weak trend with a long seasonal span. Details about the model library $\mathcal{M}$ can be found in Appendix C. Moreover, for each $m_i \in \mathcal{M}_p$, the agent generates a rationale $r_i$ explaining how data patterns in analysis report $A$ motivate the choice of $m_i$.

**Hyperparameter Optimization.** For each model $m_i \in \mathcal{M}_p$, let $\Theta_i$ denote its hyperparameter space. We sample up to $N$ configurations $\mathcal{C}_i = \{\theta_i^{(j)}\}_{j=1}^N \subseteq \Theta_i$ and evaluate each on the validation set $\tilde{\mathcal{D}}_{\text{val}}$. The optimal configuration $\theta_i^*$ is selected by minimizing validation MAPE (Mean Absolute Percentage Error):

$$\theta_i^* = \arg\min_{\theta_i \in \mathcal{C}_i} \text{MAPE}_{\text{val}}\big(m_i(\theta_i)\big), \tag{2}$$

where

$$\text{MAPE}_{\text{val}}(m_i(\theta_i)) = \frac{100\%}{|\tilde{\mathcal{D}}_{\text{val}}|} \sum_{t \in \tilde{\mathcal{D}}_{\text{val}}} \left| \frac{x_t - \hat{x}_t^{(i,\theta_i)}}{x_t} \right|. \tag{3}$$

Here $\hat{x}_t^{(i,\theta_i)}$ denotes the prediction at time step $t$ produced by model $m_i$ instantiated with hyperparameters $\theta_i$, and $x_t$ is the corresponding ground-truth value. Analogously, we also compute $\text{MAE}_{\text{val}}$ for a comprehensive performance profile, which allows for robustness checks against different error metrics. The detailed hyperparameter optimization procedure is summarized in Algorithm 1.

**Model Ranking.** After hyperparameter optimization, each candidate model $m_i$ is instantiated with its optimal configuration $\theta_i^*$ and associated validation metrics. To select a high-quality subset for ensemble construction, the $n_p$ tuned models are ranked by their validation performance. We primarily adopt validation MAPE for ranking. Specifically, the top $k$ models with the lowest validation

---

**Algorithm 1** Hyperparameter Optimization for Candidate Models

---

**Input:** Validation set $\tilde{\mathcal{D}}_{\text{val}} = \{\tilde{x}_t\}_{t=1}^{T_{val}}$, Candidate model pool $\mathcal{M}_p$
**Output:** Validation metrics $\mathcal{S}_{\text{val}}$, Optimal hyperparameter set $\Theta^*$

1: **for** $m_i \in \mathcal{M}_p$ **do**
2:     $\Theta_i \leftarrow \text{PROPOSEHYPERPARAMS}(m_i)$ # define hyperparameter space
3:     Sample $\mathcal{C}_i \sim (\Theta_i, N)$ # sample N configs from the hyperparameter space
4:     $\theta_i^* \leftarrow \arg\min_{\theta_i \in \mathcal{C}_i} \text{MAPE}_{\text{val}}\big(m_i(\theta_i), \tilde{\mathcal{D}}_{\text{val}}\big)$ # select best hyperparameters
5:     $m_i^* \leftarrow m_i(\theta_i^*)$ # instantiate tuned model
6:     $\mathcal{S}_{\text{val}}[m_i] \leftarrow \text{EVALUATE}(m_i^*, \tilde{\mathcal{D}}_{\text{val}})$ # record validation metrics
7:     $\Theta^*[m_i] \leftarrow \theta_i^*$ # record chosen hyperparameters
8: **end for**
9:
10: **return** $\mathcal{S}_{\text{val}}, \Theta^*$

---

MAPE scores are retained:

$$\mathcal{M}_{\text{selected}} = \big\{m_{(1)}(\theta_{(1)}^*), \ldots, m_{(k)}(\theta_{(k)}^*)\big\}, \quad \text{MAPE}_{\text{val}}\big(m_{(1)}\big) \leq \cdots \leq \text{MAPE}_{\text{val}}\big(m_{(k)}\big).$$

Here $m_{(j)}(\theta_{(j)}^*)$ denotes the $j$-th ranked model, ordered by ascending validation MAPE. The output of this stage is the selected models set $\mathcal{M}_{\text{selected}}$ together with tuned hyperparameters $\Theta^*$ and validation metrics $\mathcal{S}_{\text{val}}$, which serve as the foundation for ensemble construction.

### 3.4 FORECASTER

Ensemble forecasting combines complementary biases to surpass single models, cutting error under concept drift (Zhang et al., 2023b), yielding broad gains across heterogeneous patterns (Liu et al., 2025), excelling on benchmarks (Oreshkin et al., 2020), and maintaining robustness across epidemic phases (Adiga et al., 2023). Forecaster takes the top-$k$ selected models $\mathcal{M}_{\text{selected}}$ and their validation metrics $S_{\text{val}}$ as input. The agent leverages an LLM-guided policy to select an ensemble strategy from among three families: single–best selection, performance-aware averaging, or robust aggregation. The ensemble strategy and (if applicable) weights are fixed before touching the test set to avoid data leakage. With the ensemble strategy determined, Forecaster tests the ensemble model on the held-out test horizon of length $H$ to output the final forecast, and reports test metrics $S_{\text{test}}$ for comparative evaluation. This procedure balances performance and stability while attenuating outliers and regime-specific brittleness. Implementation details and ensembling rules can be found in Appendix B.

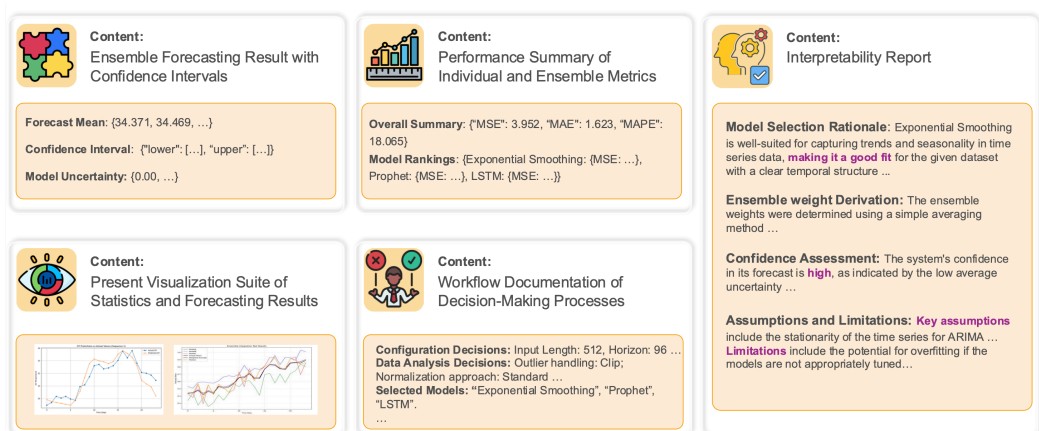

Figure 4: **Demonstration of the output comprehensive report** $\mathcal{R}$. The report consists of five parts, consolidating results, diagnostics, interpretations, and decision provenance into a transparent output.

### 3.5 REPORTER

A clear, well-structured output is essential for human scientists. Reporter outputs a comprehensive report $\mathcal{R}$ that consolidates all intermediate statistical analyses and forecasting results. Specifically, $\mathcal{R}$

includes: (1) an ensemble forecast $\hat{\mathbf{x}}_{t+1:t+H}^{\text{ens}}$ completed with confidence intervals; (2) a performance summary presenting test metrics for each model alongside the ensemble; (3) an interpretability report in which an LLM generates natural-language explanations of (i) the rationale for selecting specific models, (ii) the derivation of ensemble weights, (iii) the system's confidence in its forecast, and (iv) any underlying assumptions or limitations; (4) a visualization suite containing detailed plots for exploratory analysis and presentation; and (5) full workflow documentation that records every decision made at each phase of the pipeline. A demonstration of the generated report is in Figure 4.

The system achieves interpretability through LLM reasoning at each decision point, providing natural language explanations for model selection, hyperparameter choices, and ensemble construction strategies. This transparency enables users to understand and trust the forecasting process while benefiting from the automated optimization capabilities of the multi-agent architecture.

## 4 EXPERIMENT

In this section, we present the experiment results of TSci in comparison with both statistical and LLM-based baselines and provide a comprehensive analysis. Our framework achieves superior performance over statistical models and state-of-the-art large language models across diverse benchmarks and settings. To ensure fairness, we strictly follow the same evaluation protocols for all baselines. Unless otherwise specified, we adopt GPT-4o (Wu et al., 2023a) as the default backbone.

### 4.1 PERFORMANCE ANALYSIS

**Results.** Our brief results in Table 1 demonstrate that TSci consistently outperforms LLM-based baselines across eight benchmarks and significantly so for the majority of them. Compared with the second-best baseline, TSci reduces MAE by an average of **38.2%**. The results highlight the robustness and generalization capability of TSci across heterogeneous domains, confirming its advantage as a unified solution for time series forecasting. Figure 1a visualizes the performance comparison using min-max inversion (maps the lowest-MAE method to 100, the highest-MAE maps to 20, and others scale proportionally).

Figure 5 reports MAE on four ETT-small datasets across multiple horizons. TSci dominates statistical methods on most datasets and horizons, particularly as the forecast length increases. At short horizons, locally autoregressive structure can make simple linear models (e.g., linear regression) competitive, which match or slightly exceed TSci. But their advantage diminishes as horizon increases or patterns deviate from near-linear dynamics. The aggregate trend favors TSci, reflecting its capacity to adapt to diverse regimes while preserving short-term fidelity. Full results by datasets and horizons are provided in Appendix G.

Table 1: Time Series forecasting results compared with five LLM-based baselines. A lower value indicates better performance. Red : the best, Blue : the second best.

| Method | GPT-4o | | Gemini-2.5 Flash | | Qwen-Plus | | DeepSeek-v3 | | Claude-3.7 | | TSci (Ours) | |
|---|---|---|---|---|---|---|---|---|---|---|---|---|
| Metric | MAE | MAPE(%) | MAE | MAPE(%) | MAE | MAPE(%) | MAE | MAPE(%) | MAE | MAPE(%) | MAE | MAPE(%) |
| ETTh1 | 2.01e1 | 183.8 | 5.20 | 61.1 | 1.15e1 | 113.8 | 1.22e1 | 134.9 | 9.16 | 111.0 | **2.02** | **23.3** |
| ETTh2 | 1.82e1 | 264.6 | 1.10e1 | 81.0 | 3.27e1 | 175.6 | 2.01e1 | 121.6 | 1.16e1 | 118.6 | **4.91** | **24.7** |
| ETTm1 | 5.75 | 85.7 | 7.31 | 59.9 | 5.09 | 48.4 | 8.17 | 117.2 | 6.22 | 65.9 | **2.73** | **29.8** |
| ETTm2 | 9.94 | 50.7 | 1.60e1 | 74.7 | 1.07e1 | 71.7 | 9.01 | 39.7 | 6.94 | 41.1 | **4.87** | **31.6** |
| Weather | 6.13e1 | 10.9 | 6.52e1 | 11.8 | 4.29e1 | 6.4 | 5.20e1 | 8.3 | 4.56e1 | 6.9 | **2.91e1** | **4.4** |
| ECL | 6.33e3 | 260.2 | 8.86e2 | 45.4 | 1.66e3 | 62.9 | 68.3e3 | 235.7 | 8.44e2 | **32.2** | 6.67e2 | 40.2 |
| Exchange | 1.60e-1 | 26.2 | 1.28e-1 | 19.9 | 8.5e-2 | 13.6 | 1.75e-1 | 26.7 | 7.3e-2 | 11.8 | **4.50e-2** | **6.8** |
| ILI | 2.17e5 | 26.2 | 2.46e5 | 29.3 | 3.37e5 | 37.0 | 2.24e5 | 26.5 | 1.79e5 | 19.7 | **1.41e5** | **16.2** |
| $1^{st}$ Count | 0 | | 0 | | 0 | | 0 | | 1 | | **8** | |

### 4.2 GENERATED REPORT EVALUATION

The final comprehensive report serves as a crucial interface to access and interpret the outcomes of the framework. We evaluate the quality of the generated reports from a comprehensive perspective.

**Evaluation Metrics.** We adopt pairwise LLM-based comparison across five rubrics: AS, MJ (technical rigor), and IC, AQ, SC (communication quality). Details can be found in Appendix F. For

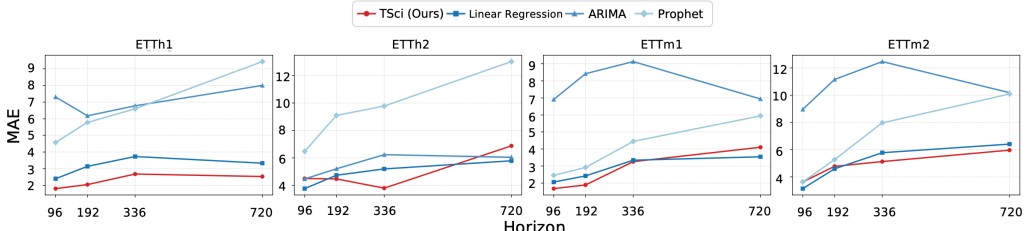

Figure 5: **Performance comparison of TSci** with statistical baselines on ETT-small benchmarks.

each rubric, we compute the *win rate*, defined as the proportion of pairwise comparisons in which our framework's report is judged superior to a baseline, excluding ties.

**Results.** As shown in Table 2, TSci consistently outperforms all baselines across the five rubrics. The largest gains appear in AS and MJ, where win rates exceed **80%** for all comparisons, underscoring the rigor and appropriateness of our analyses and model choices. Strong performance is also observed in IC and AQ (mostly above **75%**), indicating coherent reasoning and actionable recommendations. While the advantages of SC are smaller, our framework still delivers consistently structured and professional reports. Taken together, these results validate that TSci not only surpasses baselines in predictive quality, but also generates reports that are technically rigorous, interpretable, and practically useful. Figure 1b visualizes the win rate comparison (highest win rate maps to 100, the lowest to 20, and others scale linearly).

Table 2: Win rate (%) of TSci against LLM-based baselines across five rubrics.

| Baseline | AS | MJ | IC | AQ | SC |
|---|---|---|---|---|---|
| TSci *vs* GPT-4o | 80.8 | 84.6 | 80.8 | 76.9 | 71.4 |
| TSci *vs* Gemini-2.5 Flash | 81.8 | 81.8 | 63.6 | 68.2 | 53.8 |
| TSci *vs* Qwen-Plus | 83.3 | 83.3 | 79.2 | 75.0 | 75.0 |
| TSci *vs* DeepSeek-v3 | 92.3 | 84.6 | 80.8 | 76.9 | 76.9 |
| TSci *vs* Claude-3.7 | 84.7 | 87.5 | 84.6 | 80.8 | 53.8 |

### 4.3 MODEL ANALYSIS

Our results in Figure 6 indicate that ablating any of the data pre-processing, data analysis, or model optimization module degrades time-series forecasting performance.

**Effect of data preprocessing module.** Removing the data preprocessing module in Curator leads to an average of **41.80%** increase in MAE, which is the largest increase among the three modules. More specifically, the performance degeneration increases with increasing prediction horizons within one dataset. These findings demonstrate that data pre-processing contributes the most to the robustness of TSci, and underscore that cleaning, resampling, and outlier handling are crucial for analysis and especially long-horizon forecasts.

**Effect of data analysis module.** The analysis module in Curator profiles each series and serves for downstream strategies. Removing the module harms MAE of **28.3%** on average. Two minute-level cases show small improvements (ETTm1-96 and ETTm-720), suggesting minute-level data at very short/long horizons may benefit from further tuning of preprocessing and search. Overall, analysis guidance stabilizes model choice and horizon-specific settings.

**Effect of model optimization module.** The model optimization module performs parameter search for selected forecast models. Removing this module leaves a reasonable but suboptimal configuration, producing a **36.2%** MAE drop on average and a marked decline on long horizons or high-variance series where horizon chunking and window sizing matter.

### 4.4 CASE STUDY

We present a case study on the ECL dataset with horizon $H = 96$, a case where our framework surpasses other baselines by a large margin. We analyze the analysis summary generated by Curator

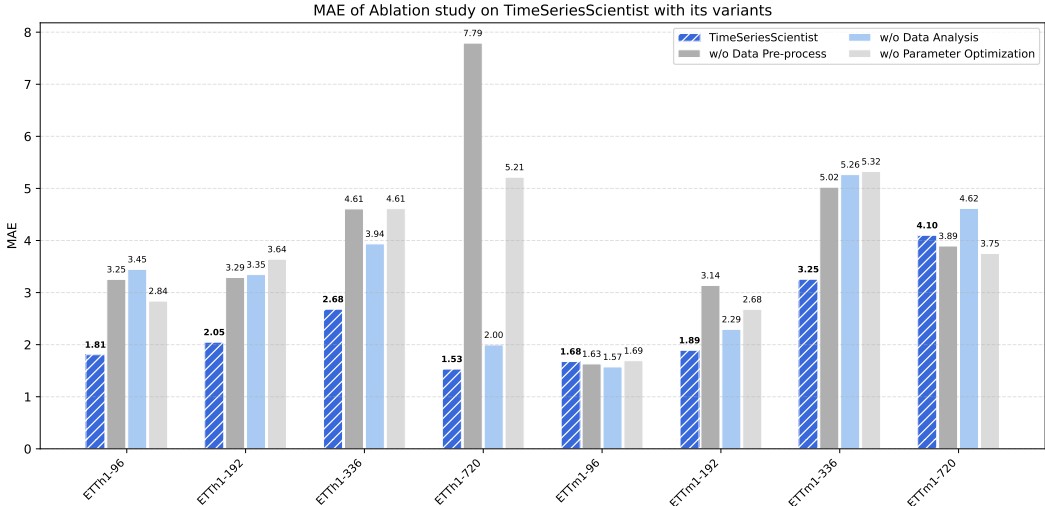

Figure 6: **Ablation study of TSci with three variants:** w/o Data Pre-process, w/o Data Analysis, and w/o Parameter Optimization. TSci attains the lowest MAE on six out of eight settings.

and the final report to highlight the effectiveness and interpretability of our agentic design. The data analysis summary, visualization, and final comprehensive report are provided in Appendix H.

The whole dataset is first divided into 25 slices, and we take one slice for study. The analysis summary in Appendix H.1 shows that the series exhibits strong cyclical fluctuations with noticeable peaks and troughs, but no persistent long-term trend. Statistical summaries indicate a symmetric distribution with light tails, as evidenced by near-zero skewness and negative kurtosis. Seasonal decomposition further confirms a strong seasonal component, while stationarity tests suggest that the data is non-stationary. Based on the analysis, Planner selected three models capable of handling non-stationary and seasonal signals, including ARIMA, Prophet, and Exponential Smoothing from the model library. The Visualization highlighted the cyclical nature of the data and irregular spikes, reinforcing the importance of models that adapt to seasonality. Following this, Forecaster produced ensemble forecasts and assigned higher weights to models capturing seasonal dynamics.

Figure 14 shows the ensemble forecast with individual model predictions. While individual models such as ARIMA and Prophet struggled with accumulated errors over the horizon $H = 96$, our ensemble remained stable and aligned with the seasonal cycles. The ensemble strategy given by the LLM mitigates errors from the individual model and produces a more stable forecast. The final comprehensive report further provided human-readable explanations, linking the model choices directly to the identified seasonality and non-stationarity in the data.

This case study demonstrates that our framework is not only more accurate than baselines but also produces interpretable outputs. The generated reports bridge the gap between automated forecasting and human reasoning by explaining why certain models are preferred, how data characteristics influence forecasts, and where potential risks (e.g., non-stationarity, irregular spikes) lie.

## 5 Conclusions and Future Work

We introduced TimeSeriesScientist, the first end-to-end, agentic framework that automates univariate time series forecasting via LLM reasoning. Extensive experiments across diverse benchmarks show consistent gains over state-of-the-art LLM baselines, demonstrating both prediction accuracy and report interpretability. This work provides the first step toward a unified, domain-agnostic approach for univariate time series forecasting, bridging the gap between traditional forecasting methods and the emerging capabilities of foundation models. Future directions include extending to multimodal settings for broader applicability and incorporating external knowledge and efficiency-oriented designs to enhance interpretability and scalability. We hope this work inspires further research at the intersection of time series forecasting, agentic reasoning, and foundation models.

## 6 REPRODUCIBILITY STATEMENT

We have taken several steps to ensure the reproducibility of our work. All datasets used in this work are publicly available, and we provide a complete description of the datasets in Appendix D.2. The implementation details of our proposed framework, including model configurations, training protocols, and evaluation metrics, are described in Section 4, with further hyperparameter settings reported in Appendix D. To facilitate replication, we release the source code and experiment scripts in an anonymous repository as supplementary material. Additional information is also included in the appendix. Together, these resources ensure that our results can be independently verified and extended.

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

# A DATA PROCESSING STRATEGIES

We formalize a leakage-safe toolkit for detecting and repairing data issues in time series $\{x_t\}_{t=1}^T$. All statistics are estimated on *rolling (local)* windows to accommodate non-stationarity. Let $\mathcal{O}$ and $\mathcal{M}$ denote the sets of outlier and missing indices, respectively. The agent reasons on data statistics and

## A.1 OUTLIER DETECTION

**Rolling IQR.** On a window $\mathcal{W}_t$ of length $w$, compute its first and third quantile:

$$Q_1(\mathcal{W}_t),\ Q_3(\mathcal{W}_t),\quad \mathrm{IQR}_t = Q_3 - Q_1. \tag{4}$$

The outlier criterion:

$$x_t \text{ is outlier if}\quad x_t < Q_1 - \alpha \cdot \mathrm{IQR}_t\ \text{ or }\ x_t > Q_3 + \alpha \cdot \mathrm{IQR}_t, \tag{5}$$

with a common choice $\alpha=1.5$. If strong seasonality exists, set $w$ to one or two seasonal cycles.

**Rolling Z-Score.** Estimate $\mu_t, \sigma_t$ within window $\mathcal{W}_t$ and define

$$z_t = \frac{|x_t - \mu_t|}{\sigma_t},\qquad x_t \text{ is outlier if } z_t > \alpha, \tag{6}$$

typically $\alpha \in [3, 4]$ for online detection. For skewed/heavy-tailed data, replace $\mu_t$ and $\sigma_t$ by the median and MAD:

$$\mu_t \leftarrow \mathrm{median}(\mathcal{W}_t),\qquad \sigma_t \leftarrow 1.4826 \cdot \mathrm{MAD}(\mathcal{W}_t), \tag{7}$$

then apply the same threshold on $z_t$.

**Percentile Rule.** Using empirical quantiles within $\mathcal{W}_t$ (adaptive) or from the training segment (frozen),

$$x_t \text{ is outlier if}\quad x_t < P_{\mathrm{lower}}\ \text{ or }\ x_t > P_{\mathrm{upper}}, \tag{8}$$

e.g., $(P_{\mathrm{lower}}, P_{\mathrm{upper}}) = (1\%, 99\%)$ or $(0.5\%, 99.5\%)$.

## A.2 OUTLIER HANDLING

**Clipping / Winsorization.** Let $L$ and $U$ be lower/upper bounds from non-outliers (or from quantiles such as $P_{1\%}, P_{99\%}$):

$$x_t^{\mathrm{clean}} = \begin{cases} L, & x_t < L, \\ U, & x_t > U, \\ x_t, & \text{otherwise.} \end{cases} \tag{9}$$

**Interpolation (Segment-Aware).** For a contiguous outlier segment $t \in [a, b]$ with nearest clean neighbors $\tau_0 < a$ and $\tau_1 > b$,

$$x_t^{\mathrm{clean}} = x_{\tau_0} + \frac{t - \tau_0}{\tau_1 - \tau_0}\big(x_{\tau_1} - x_{\tau_0}\big),\qquad t = a, \ldots, b. \tag{10}$$

For isolated points, this reduces to the two-point linear case ($x_t^{\mathrm{clean}} = \frac{x_{t-1}+x_{t+1}}{2}$).

**Forward/Backward Fill.** Short gaps in level-like processes:

$$x_t^{\mathrm{clean}} = x_{t-1}\ \text{(FFill)},\qquad x_t^{\mathrm{clean}} = x_{t+1}\ \text{(BFill)}. \tag{11}$$

**Local Mean/Median Replacement.** Within a causal neighborhood $\mathcal{N}_t$ (e.g., last $w$ points),

$$x_t^{\mathrm{clean}} = \frac{1}{|\mathcal{N}_t|} \sum_{i \in \mathcal{N}_t} x_i\quad \text{or}\quad x_t^{\mathrm{clean}} = \mathrm{median}\{x_i : i \in \mathcal{N}_t\}. \tag{12}$$

Median is preferred under heavy tails or residual outliers.

**Light Causal Smoothing.** After replacement, apply a causal moving average to suppress residual spikes:

$$x_t^{\text{clean}} = \frac{1}{w} \sum_{i=0}^{w-1} x_{t-i}. \tag{13}$$

Use small $w$ to limit lag and peak attenuation.

### A.3 MISSING-VALUE HANDLING

**Linear Interpolation (Segment-Aware).** For a missing segment $t \in [a, b]$ bounded by clean points $\tau_0 < a$ and $\tau_1 > b$,

$$x_t = x_{\tau_0} + \frac{t - \tau_0}{\tau_1 - \tau_0}\big(x_{\tau_1} - x_{\tau_0}\big), \qquad t = a, \ldots, b. \tag{14}$$

**Forward/Backward Fill.**

$$x_t = x_{t-1} \ \ (\text{FFill}), \qquad x_t = x_{t+1} \ \ (\text{BFill}). \tag{15}$$

**Local Mean/Median Fill.** Estimate within a local window (prefer causal in evaluation):

$$x_t = \frac{1}{n} \sum_{i=1}^{n} x_i \quad \text{or} \quad x_t = \text{median}\{x_1, \ldots, x_n\}. \tag{16}$$

**Zero Fill (Semantic Zero Only).**

$$x_t = 0, \tag{17}$$

used only when zero has a clear meaning (e.g., counts/absence).

## B ENSEMBLE

Here we introduce the detailed

**Setup.** Let $\mathcal{M}_{\text{selected}} = \{m_i(\theta_i^*)\}_{i=1}^{k}$ be the top-$k$ models returned by Planner with tuned hyperparameters $\theta_i^*$ and validation scores $S_{\text{val}}$. For each model $m_i$, we compute a scalar validation loss $s_i$ (lower is better) by aggregating the normalized metric vector $\boldsymbol{\ell}_i \in \mathbb{R}^M$ (e.g., MAE, MAPE):

$$s_i = \sum_{m=1}^{M} \alpha_m \, \text{norm}(\ell_{i,m}), \quad \alpha_m \geq 0, \ \sum_m \alpha_m = 1. \tag{18}$$

On the test horizon of length $H$, model $m_i$ outputs $\hat{x}_{1:H}^{(i)}$. An ensemble produces $\hat{x}_h = \sum_{i=1}^{k} w_i \, \hat{x}_h^{(i)}$ with horizon-wise fixed weights $w_i \geq 0$, $\sum_i w_i = 1$. All choices below depend only on $S_{\text{val}}$ and pre-specified hyperparameters; no test data is touched.

**(A) Single–Best Selection.** Pick the model with the best validation score and use it alone:

$$i^\star = \arg\min_{i \in [k]} s_i, \qquad w_{i^\star} = 1, \ w_{j \neq i^\star} = 0. \tag{19}$$

*When used.* Prefer (19) if the leader is clearly ahead:

$$\text{gap} = \frac{s_{(2)} - s_{(1)}}{s_{(1)}} \geq \delta, \quad \text{with } s_{(1)} \leq s_{(2)} \leq \cdots \leq s_{(k)}, \tag{20}$$

where $\delta$ is a small margin (default $\delta = 0.05$). This avoids diluting a dominant model with weaker ones.

**(B) Performance-Aware Averaging.** Assign higher weights to better validation performance while preventing over-concentration. We use a temperatured inverse-loss scheme with shrinkage:

$$\tilde{w}_i = (s_i + \varepsilon)^{-\beta}, \qquad \beta > 0, \ \varepsilon > 0, \tag{21}$$

$$w_i^{\text{perf}} = \frac{\exp\big(-\log \tilde{w}_i / \tau\big)}{\sum_{j=1}^{k} \exp\big(-\log \tilde{w}_j / \tau\big)} = \frac{\tilde{w}_i^{1/\tau}}{\sum_{j=1}^{k} \tilde{w}_j^{1/\tau}}, \tag{22}$$

$$w_i = (1 - \lambda) \, \text{clip}\big(w_i^{\text{perf}}, w_{\min}, w_{\max}\big) + \lambda \cdot \frac{1}{k}, \tag{23}$$

with defaults $\beta{=}1$, $\tau{=}1$, $\lambda{=}0.1$, $w_{\min}{=}0.02$, $w_{\max}{=}0.80$, and $\varepsilon{=}10^{-8}$. When multiple metrics are used, $s_i$ comes from (18) with min–max normalization inside $\text{norm}(\cdot)$ across the $k$ candidates. The shrinkage in (23) stabilizes weights in small-$k$ regimes and under close scores.

**(C) Robust Aggregation.** When candidate predictions disagree substantially, use distribution-robust, order-statistic based pooling at each horizon index $h$:

$$\text{Median:} \quad \hat{x}_h^{\text{med}} = \text{median}\big\{\hat{x}_h^{(1)}, \ldots, \hat{x}_h^{(k)}\big\}, \tag{24}$$

$$\text{Trimmed mean:} \quad \hat{x}_h^{\text{trim}} = \frac{1}{k - 2\lfloor \rho k \rfloor} \sum_{i=\lfloor \rho k \rfloor + 1}^{k - \lfloor \rho k \rfloor} \hat{x}_{h:\uparrow}^{(i)}, \tag{25}$$

where $\hat{x}_{h:\uparrow}^{(i)}$ denotes the $i$-th smallest prediction at step $h$ and $\rho \in [0, 0.25)$ is the trimming fraction (default $\rho = 0.1$). Median (24) has a $50\%$ breakdown point; the trimmed mean (25) trades slightly lower robustness for variance reduction.

**Notes on implementation.** (i) Weights $w_i$ are horizon-wise constant to avoid step-wise overfitting; (ii) when Curator applies scaling (e.g., z-score), ensembling is performed in the scaled space and then inverted; (iii) performance aggregation (18) can emphasize a primary metric by setting its $\alpha_m$ larger (we use $\alpha_{\text{MAE}}{=}\alpha_{\text{MAPE}}{=}0.5$ by default); (iv) computational cost is $O(kH)$ for all strategies; (v) for $k{=}1$, (19) is used by definition.

## C  MODEL LIBRARY

Here is a full list of time series models that we implement. The 21 models can be divided into 5 categories: 1) Traditional Statistical models; 2) Regression-based machine learning (ML) models; 3) Tree-based Models (Ensemble method); 4) Neural Network Models (Deep Learning); 5) Specialized Time Series Models. Details are listed in Table 3.

## D  EXPERIMENTAL DETAILS

### D.1  IMPLEMENTATIONS

We use OpenAI GPT-4o (OpenAI et al., 2024) as the default backbone model. Due to a limited budget, we divided all datasets into 25 slices and conducted experiments on these slices instead of the entire dataset. The input time series length $T$ for each slice is set as 512, and we use four different prediction horizons $H \in \{96, 192, 336, 720\}$. The evaluation metrics include mean absolute error (MAE) and mean absolute percentage error (MAPE). We report the averaged results from the 25 slices.

### D.2  DATASET DETAILS

Dataset statistics are summarized in Table 4. We evaluate the univariate time series forecasting performance on the well-established eight different benchmarks, including four ETT datasets, Weather, Electricity, Exchange, and ILI from Wu et al. (2023a).

Table 3: Implemented time series forecasting model library in `model_library.py`.

| Category | Model | Function name |
|---|---|---|
| Statistical (7) | ARIMA | `predict_arima` |
| | RandomWalk | `predict_random_walk` |
| | ExponentialSmoothing | `predict_exponential_smoothing` |
| | MovingAverage | `predict_moving_average` |
| | TBATS | `predict_tbats` |
| | Theta | `predict_theta` |
| | Croston | `predict_croston` |
| ML regression (6) | LinearRegression | `predict_linear_regression` |
| | PolynomialRegression | `predict_polynomial_regression` |
| | RidgeRegression | `predict_ridge_regression` |
| | LassoRegression | `predict_lasso_regression` |
| | ElasticNet | `predict_elastic_net` |
| | SVR | `predict_svr` |
| Tree-based (4) | RandomForest | `predict_random_forest` |
| | GradientBoosting | `predict_gradient_boosting` |
| | XGBoost | `predict_xgboost` |
| | LightGBM | `predict_lightgbm` |
| Neural networks (2) | NeuralNetwork | `predict_neural_network` |
| | LSTM | `predict_lstm` |
| Specialized (2) | Prophet | `predict_prophet` |
| | Transformer | `predict_transformer` |

Table 4: Summary of datasets across different domains.

| Dataset | Domain | Length | Frequency | Duration |
|---|---|---|---|---|
| ETTh1, ETTh2 | Electricity | 17,420 | 1 hour | 2016.07.01 - 2018.06.26 |
| ETTm1, ETTm2 | Electricity | 69,680 | 15 mins | 2016.07.01 - 2018.06.26 |
| Weather | Environment | 52,696 | 10 mins | 2020.01.01 - 2021.01.01 |
| Electricity | Electricity | 26,304 | 1 hour | 2016.07.01 - 2019.07.02 |
| Exchange | Economic | 7,588 | 1 day | 1990.01.01 - 2010.10.10 |
| ILI | Health | 966 | 1 week | 2002.01.01 - 2020.06.30 |

## D.3 BASELINES

We benchmark TSci against several leading large language models, including GPT-4o, Gemini-2.5 Flash (Gemini Team, Google, 2025), Qwen-Plus (Cloud, 2025a), DeepSeek-v3 (Liu et al., 2024), and Claude-3.7 (Cloud, 2025b).

## E VISUALIZATIONS

### E.1 LLM GUIDED DATA VISUALIZATIONS

Our framework generates comprehensive visualizations during the pre-processing stage to facilitate data understanding and quality assessment. The visualization pipeline employs a multi-panel approach to systematically examine time series characteristics.

**Time Series Overview Plot.** The primary visualization component displays the raw time series data with temporal indexing on the x-axis and corresponding values on the y-axis. This panel serves as the foundational view for identifying global patterns, potential anomalies, and overall data structure.

The visualization incorporates grid lines with reduced opacity ($\alpha = 0.3$) to enhance readability while maintaining focus on the data trajectory, as shown in Figure 7.

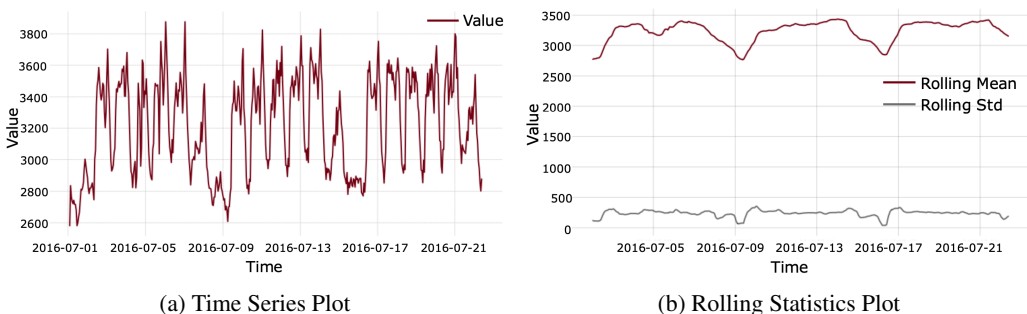

    (a) Time Series Plot                 (b) Rolling Statistics Plot

Figure 7: **Example of time series overview plot** on one slice of ECL dataset with input length $T = 512$. Figure 7a displays the raw data. Figure 7b shows the rolling mean and rolling standard deviation of the data slice.

**Time Series Decomposition Analysis Plot.** To comprehensively understand the underlying structure of the time series data, we employ seasonal decomposition to decompose the original series into four interpretable components, as shown in Figure 8. The decomposition follows the additive model $X_t = T_t + S_t + R_t$, where $X_t$ represents the original observed values, $T_t$ denotes the trend component capturing long-term systematic changes, $S_t$ indicates the seasonal component revealing periodic patterns with a fixed frequency, and $R_t$ represents the residual component containing random noise and unexplained variations. The trend component helps identify the overall direction and magnitude of change over time, while the seasonal component exposes recurring patterns that may be crucial for forecasting accuracy. The residual component serves as a diagnostic tool to assess the adequacy of the decomposition and identify potential anomalies or structural breaks. This four-panel visualization provides essential insights for selecting appropriate preprocessing strategies and forecasting models, as the presence of strong trends or seasonality directly informs the choice of detrending methods and seasonal adjustment techniques.

**Autocorrelation Analysis Plot.** To assess the temporal dependencies and identify potential patterns in the time series data, we employ the autocorrelation function (ACF) and partial autocorrelation function (PACF) plots, as shown in Figure 9. The ACF measures the linear relationship between observations at different time lags, revealing the overall memory structure and helping identify seasonal patterns, trends, and the presence of unit roots. The PACF, on the other hand, measures the correlation between observations at a specific lag while controlling for the effects of intermediate lags, providing insights into the optimal order of autoregressive models and helping distinguish between autoregressive and moving average components. These diagnostic plots are essential for model identification in ARIMA modeling, as they reveal the underlying stochastic process characteristics and guide the selection of appropriate differencing operations and model parameters. The ACF and PACF analysis enables us to understand the temporal structure of the data, identify potential non-stationarity issues, and inform the choice of appropriate forecasting models based on the observed correlation patterns.

### E.2 TECHNICAL IMPLEMENTATION DETAILS

All visualizations are generated using Matplotlib and seaborn libraries with consistent styling parameters to ensure reproducibility and professional presentation. The time series plots employ a line width of 2.0 pixels with a standardized color palette (#c83e4b for primary series), while distribution plots utilize a 2×2 subplot layout combining time series visualization, histogram with kernel density estimation (KDE), box plots, and Q-Q plots for comprehensive distributional analysis. Rolling statistics plots compute moving averages and standard deviations using configurable window sizes (default 24 periods) with distinct color coding for trend and volatility components. Seasonal decomposition leverages the statsmodels.tsa.seasonal.seasonal_decompose function with additive decomposition and configurable seasonal periods, while autocorrelation analysis employs plot_acf and plot_pacf functions with 40-lag windows for optimal model identification. All plots feature white

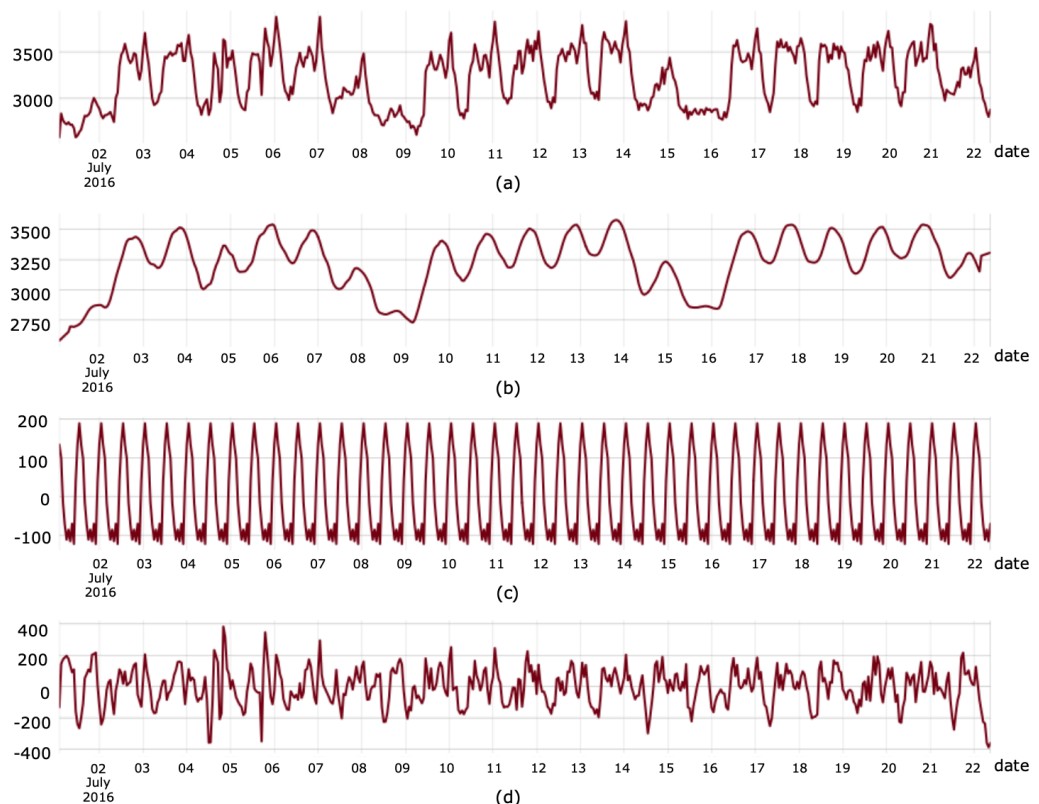

Figure 8: **Example of time series decomposition analysis plot** on ECL dataset with input length $T$=512. Figure 7(a) is the plot of the original time series $X_t$. Figure 7(b) is the plot of the trend $T_t$. Figure 7(c) is the plot of the seasonal component $S_t$. Figure 7(d) is the plot of the residual component $R_t$.

backgrounds with black grid lines (major grid: solid lines, 0.5px width, 30% opacity; minor grid: dotted lines, 0.3px width, 20% opacity) and are saved as high-resolution PDF files (300 DPI) with tight bounding boxes to ensure publication-quality output. The visualization generation process is fully automated through LLM-driven configuration, allowing dynamic adaptation of plot parameters based on data characteristics and analysis requirements.

### E.3 OUTPUT AND INTEGRATION

The visualization pipeline generates standardized output files in PDF format, with configurable save paths and automatic directory creation. Each visualization includes comprehensive logging for audit trails and debugging purposes. The system integrates seamlessly with the broader time series prediction framework, automatically generating visualizations during the pre-processing stage and storing them for subsequent analysis and reporting phases.

These pre-processing visualizations serve as the foundation for data-driven decision making, enabling researchers and practitioners to understand their time series data characteristics before proceeding to model selection and forecasting stages.

## F REPORT EVALUATION RUBRICS

Here, we describe the details of the five rubrics that comprehensively evaluate the generated report:

**Analysis Soundness (AS):** Evaluates the rigor and correctness of exploratory data analysis, including the handling of missing values, anomaly detection, and identification of seasonality or trends.

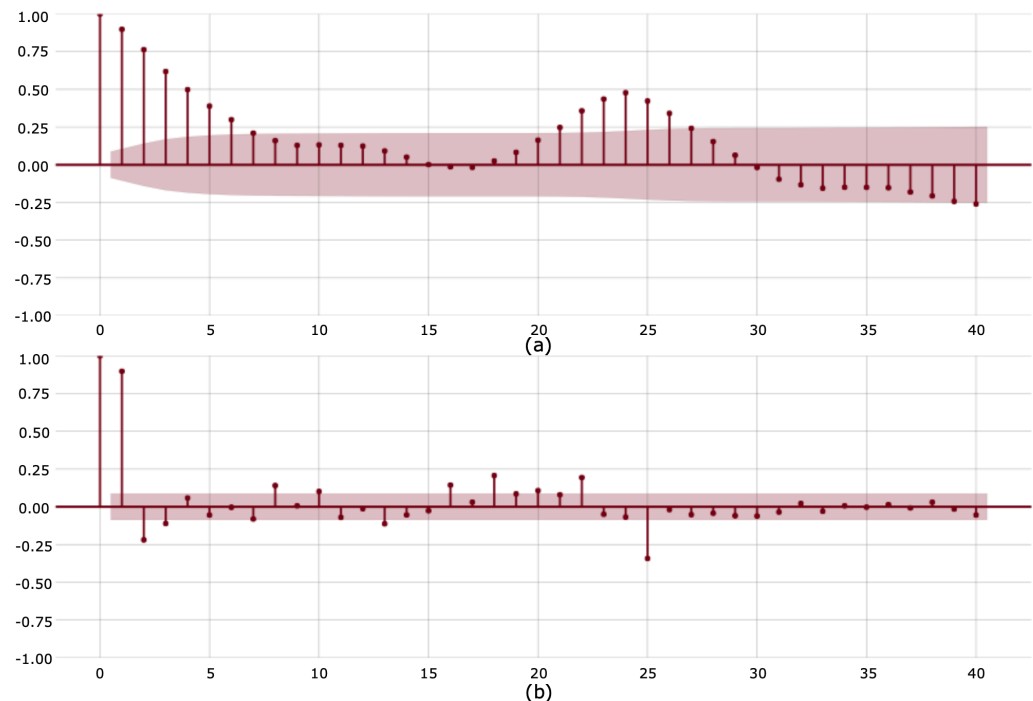

Figure 9: **Example of autocorrelation analysis plot** on ECL dataset with input length $T = 512$. Figure 8(a) is the ACF plot, and Figure 8(b) is the PACF plot.

**Model Justification (MJ):** Assesses whether the chosen forecasting models are appropriate for the data characteristics and whether the selection is supported by clear, evidence-based justification.

**Interpretive Coherence (IC):** Measures the logical consistency and alignment of the report's reasoning, ensuring interpretations of diagnostics, errors, and results form a coherent narrative.

**Actionability Quotient (AQ):** Judges the extent to which the report provides concrete, evidence-backed, and practically useful recommendations for decision making or system improvement.

**Structural Clarity (SC):** Examines the organization, readability, and professionalism of the report, including section structure, flow, and correct referencing of figures and tables.

The five rubrics comprehensively evaluate the generated report along two dimensions: **AS** and **MJ** assess the technical rigor of analysis and modeling choices, while **IC**, **AQ**, and **SC** assess the communication quality and practical usefulness of the report. For each rubric, we compute the *win rate*, defined as the proportion of pairwise comparisons in which our framework's report is judged superior to a baseline, excluding ties.

## G FULL EXPERIMENT RESULTS

Here we present the full experiment results of our TSci on eight datasets against five LLM-based baselines, as shown in Table 5 and Table 6. $1^{st}$ Count row at the end of Table 6 indicates the number of test cases where the model achieves the best performance across all datasets. TSci achieves superior performance across the majority of datasets and forecasting horizons (Figure 10), demonstrating its LLM-driven reasoning capacity in time series forecasting. Figure 11 shows the complete result of TSci compared with three statistical baselines on eight datasets. Figure 12 and 13 show the MAE and MAPE distribution across datasets and horizons.

Table 5: Time series forecasting results. A lower value indicates better performance. **Red** : the best, Blue : the second best.

| Methods | GPT-4o | | Gemini-2.5 Flash | | Qwen-Plus | | DeepSeek-v3 | | Claude-3.7 | | TSci (Ours) | |
|---|---|---|---|---|---|---|---|---|---|---|---|---|
| Metric | MAE | MAPE | MAE | MAPE | MAE | MAPE | MAE | MAPE | MAE | MAPE | MAE | MAPE |
| ETTh1 96 | 6.39 | 69.3 | 4.99 | 71.8 | 6.50 | 74.8 | 7.24 | 90.9 | 5.58 | 75.3 | **1.81** | **13.9** |
| 192 | 1.10e1 | 89.9 | 5.04 | 57.4 | 9.79 | 108.3 | 8.60 | 104.8 | 5.99 | 82.9 | **2.05** | **31.0** |
| 336 | 2.18e1 | 319.2 | 5.29 | 51.6 | 1.30e1 | 143.0 | 1.55e1 | 198.4 | 7.99 | 124.8 | **2.68** | **31.7** |
| 720 | 4.14e1 | 256.9 | 5.46 | 63.5 | 1.67e1 | 129.2 | 1.76e1 | 145.6 | 1.71e1 | 161.0 | **1.53** | **16.7** |
| Avg | 2.01e1 | 183.8 | 5.20 | 61.1 | 1.15e1 | 113.8 | 1.22e1 | 134.9 | 9.16 | 111.0 | **2.02** | **23.3** |
| ETTh2 96 | 1.09e1 | 190.9 | 1.16e1 | 74.7 | 3.34e1 | 320.1 | 1.02e1 | 47.2 | 8.56 | 202.7 | **4.50** | **18.9** |
| 192 | 1.45e1 | 304.2 | 1.30e1 | 102.3 | 1.65e1 | 118.7 | 1.27e1 | 147.2 | 9.62 | 107.0 | **4.47** | **12.8** |
| 336 | 2.21e1 | 441.2 | 8.76 | 65.5 | 2.49e1 | 74.5 | 1.62e1 | 118.6 | 9.95 | 70.4 | **3.81** | **10.7** |
| 720 | 2.53e1 | 121.9 | 1.08e1 | 81.6 | 5.58e1 | 189.0 | 4.13e1 | 173.4 | 1.82e1 | 94.0 | **6.88** | **56.2** |
| Avg | 1.82e1 | 264.6 | 1.10e1 | 81.0 | 3.27e1 | 175.6 | 2.01e1 | 121.6 | 1.16e1 | 118.5 | **4.91** | **24.7** |
| ETTm1 96 | 2.68 | 24.3 | 5.91 | 43.0 | 4.01 | 43.9 | 3.53 | 31.7 | 3.09 | 26.8 | **1.68** | **15.7** |
| 192 | 5.84 | 78.8 | 8.21 | 56.1 | 5.56 | 67.4 | 7.94 | 91.4 | 5.80 | 52.0 | **1.89** | **19.9** |
| 336 | 6.86 | 147.9 | 8.06 | 61.5 | 8.48 | 70.6 | 1.23e1 | 206.1 | 8.23 | 67.1 | **3.26** | **31.5** |
| 720 | 7.62 | 91.7 | 7.04 | 79.0 | 2.31 | 11.9 | 8.97 | 139.4 | 7.78 | 117.9 | **4.10** | **52.0** |
| Avg | 5.75 | 85.7 | 7.31 | 59.9 | 5.09 | 48.4 | 8.17 | 117.1 | 6.22 | 65.9 | **2.73** | **29.8** |
| ETTm2 96 | 5.52 | 29.6 | 1.30e1 | 58.2 | 7.84 | 109.2 | 4.81 | **20.7** | 4.35 | 47.8 | **3.63** | 40.5 |
| 192 | 9.22 | 43.2 | 1.41e1 | 58.7 | 7.24 | **28.6** | 7.06 | 35.2 | 9.08 | 39.1 | **4.77** | 30.5 |
| 336 | 1.11e1 | 61.5 | 1.33e1 | 78.6 | 1.18e1 | 46.0 | 1.09e1 | 44.9 | 7.97 | 34.3 | **5.12** | **27.6** |
| 720 | 1.39e1 | 68.4 | 2.34e1 | 103.4 | 1.57e1 | 102.9 | 1.33e1 | 57.9 | 6.38 | 43.0 | **5.96** | **27.6** |
| Avg | 9.94 | 50.7 | 1.60e1 | 74.7 | 1.07e1 | 71.7 | 9.01 | 39.7 | 6.94 | 41.1 | **4.87** | **31.6** |

Table 6: Time series forecasting results (continuing). A lower value indicates better performance. **Red** : the best, Blue : the second best.

| Methods | GPT-4o | | Gemini-2.5 Flash | | Qwen-Plus | | DeepSeek-v3 | | Claude-3.7 | | TSci (Ours) | |
|---|---|---|---|---|---|---|---|---|---|---|---|---|
| Metric | MAE | MAPE | MAE | MAPE | MAE | MAPE | MAE | MAPE | MAE | MAPE | MAE | MAPE |
| Weather 96 | 2.16e1 | 5.1 | 6.59e1 | 15.5 | 2.25e1 | 5.2 | **1.54e1** | **3.6** | 1.83e1 | 4.3 | 1.63e1 | 3.8 |
| 192 | 4.07e1 | 9.5 | 3.84e1 | 9.0 | 3.17e1 | 7.4 | 2.84e1 | 6.6 | 3.97e1 | 9.3 | **1.60e1** | **3.6** |
| 336 | 6.89e1 | 6.6 | **5.92e1** | **4.4** | 6.96e1 | 6.4 | 8.06e1 | 8.5 | 7.24e1 | 6.6 | 6.13e1 | 5.0 |
| 720 | 1.14e2 | 22.4 | 9.74e1 | 18.5 | 4.79e1 | 6.6 | 8.37e1 | 14.4 | 5.19e1 | 7.5 | **2.29e1** | **5.1** |
| Avg | 6.13e1 | 10.9 | 6.52e1 | 11.8 | 4.29e1 | 6.4 | 5.20e1 | 8.3 | 4.56e1 | 6.9 | **2.91e1** | **4.4** |
| ECL 96 | 2.09e3 | 63.6 | 7.37e2 | 22.8 | 1.09e3 | 32.6 | 1.36e3 | 42.2 | 8.21e2 | 23.9 | **3.94e2** | **11.2** |
| 192 | 3.64e3 | 109.0 | 1.35e3 | 41.1 | 1.42e3 | 42.6 | 2.06e3 | 62.1 | 5.05e2 | 15.2 | **4.50e2** | **13.6** |
| 336 | 5.85e3 | 252.3 | **7.79e2** | **63.5** | 1.18e3 | 72.6 | 4.89e3 | 182.1 | 1.26e3 | 39.8 | 9.68e2 | 77.3 |
| 720 | 1.38e4 | 615.9 | **6.75e2** | **54.2** | 2.97e3 | 103.8 | 1.90e4 | 656.5 | 7.93e2 | 50.0 | 8.56e2 | 58.8 |
| Avg | 6.33e3 | 260.2 | 8.86e2 | 45.4 | 1.66e3 | 62.9 | 6.83e3 | 235.7 | 8.44e2 | 32.2 | **6.67e2** | 40.2 |
| Exchange 96 | 6.21e-2 | 9.5 | 5.46e-2 | 8.8 | 3.21e-2 | 5.1 | 5.46e-2 | 8.3 | 3.08e-2 | 4.8 | **2.46e-2** | **3.8** |
| 192 | 1.09e-1 | 17.6 | 2.34e-1 | 35.3 | 6.04e-2 | 10.2 | 8.40e-2 | 13.5 | 8.75e-2 | 14.6 | **3.85e-2** | **5.8** |
| 336 | 1.52e-1 | 26.0 | 1.06e-1 | 17.1 | 7.96e-2 | 12.3 | 2.25e-1 | 35.9 | 6.65e-2 | 10.8 | **5.76e-2** | **8.9** |
| 720 | 3.14e-1 | 51.9 | 1.15e-1 | 18.4 | 1.70e-1 | 26.9 | 3.37e-1 | 49.1 | 1.06e-1 | 17.1 | **5.76e-2** | **8.8** |
| Avg | 1.60e-1 | 26.2 | 1.28e-1 | 19.9 | 8.50e-2 | 13.6 | 1.75e-1 | 26.7 | 7.30e-2 | 11.8 | **4.50e-2** | **6.8** |
| ILI 24 | 1.58e5 | 18.4 | 2.48e5 | 28.5 | 3.49e5 | 38.9 | 1.86e5 | 21.6 | 1.56e5 | 17.4 | **1.41e5** | **16.5** |
| 36 | 1.93e5 | 24.0 | 2.53e5 | 32.0 | 3.05e5 | 32.5 | 1.92e5 | 22.8 | 1.86e5 | 20.3 | **1.48e5** | **16.8** |
| 48 | 2.45e5 | 28.9 | 2.83e5 | 34.1 | 3.67e5 | 41.3 | 2.58e5 | 30.5 | 1.76e5 | 19.0 | **1.34e5** | **15.6** |
| 60 | 2.72e5 | 33.4 | 1.98e5 | 22.6 | 3.29e5 | 35.5 | 2.62e5 | 31.3 | 1.98e5 | 22.1 | **1.40e5** | **16.2** |
| Avg | 2.17e5 | 26.2 | 2.46e5 | 29.3 | 3.37e5 | 37.1 | 2.24e5 | 26.5 | 1.79e5 | 19.7 | **1.41e5** | **16.3** |
| $1^{st}$ Count | 0 | | 3 | | 2 | | 2 | | 3 | | 35 | |

# H CASE STUDY ON ECL DATASET

## H.1 ANALYSIS SUMMARY

This analysis summary presents the findings from a time series forecasting experiment conducted on the ECL dataset. The analysis focused on understanding the trend, seasonality, and stationarity of the data, and potential improvements for future forecasting efforts.

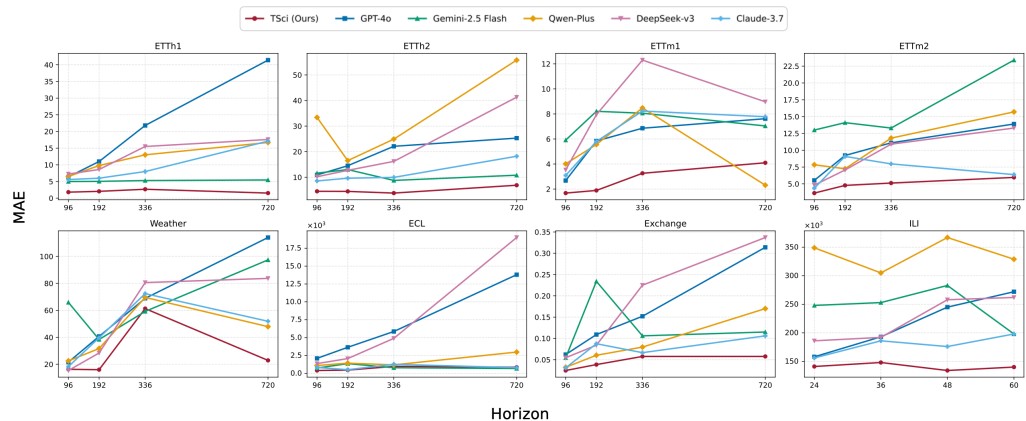

Figure 10: **Performance comparison of TSci** with five LLM-based baselines across eight datasets.

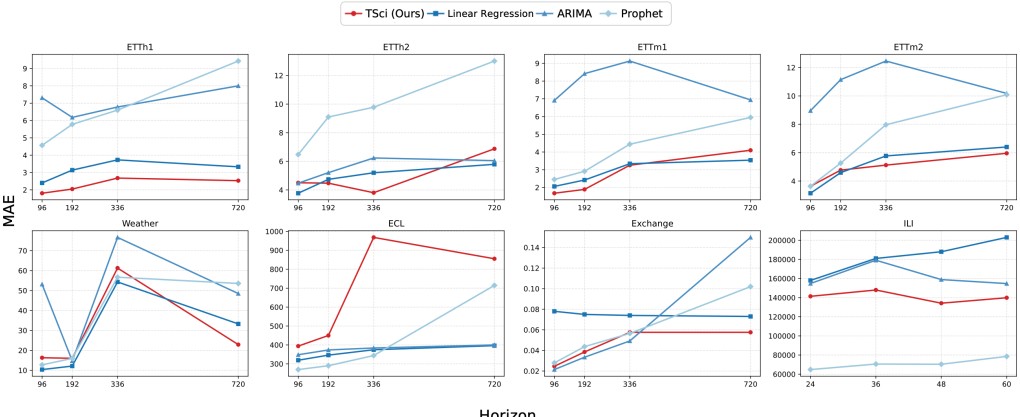

Figure 11: **Performance comparison of TSci** with three statistical baselines across eight datasets.

Analysis Summary generated from Curator

"Trend Analysis": {
"Description": "The time series data exhibits an overall upward trend over the observed period. This is evident from the increasing values towards the end of the series compared to the beginning. The trend is not strictly linear, as there are fluctuations and periods of decline, but the general direction is upward.",
"Strength": "Moderate to strong, with noticeable periods of growth, particularly towards the latter part of the series.",
"Supporting Evidence": "The Basic Time Series Plot and Rolling Statistics visualizations show a clear upward trajectory, especially in the latter half of the data."
},
"Seasonality Analysis": {
"Description": "There are indications of seasonality in the data, with recurring patterns that suggest periodic fluctuations. These patterns are not strictly regular but suggest some form of cyclical behavior.",
"Periodicity": "The exact periodicity is not immediately clear from the data provided, but the Seasonal Decomposition visualization should provide insights into the seasonal component.",
"Supporting Evidence": "The Seasonal Decomposition visualization will help identify and confirm any seasonal patterns present in the data."

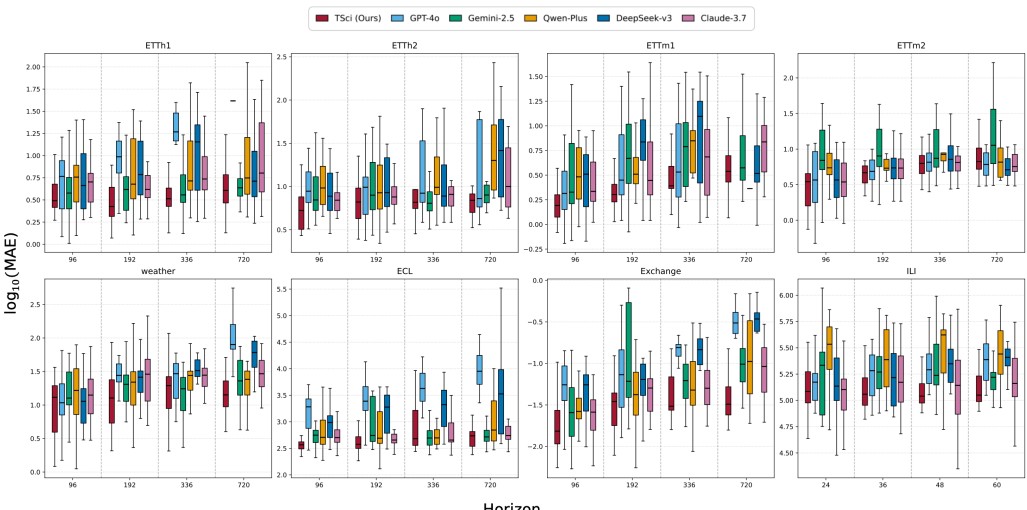

Figure 12: **Slice-level MAE distributions across datasets and horizons.** The 2×4 grid organizes subplots by dataset; within each subplot, four horizons are separated by dashed lines, and six methods are shown as grouped boxplots. Y-axis uses $log_{10}$ scale; lower is better.

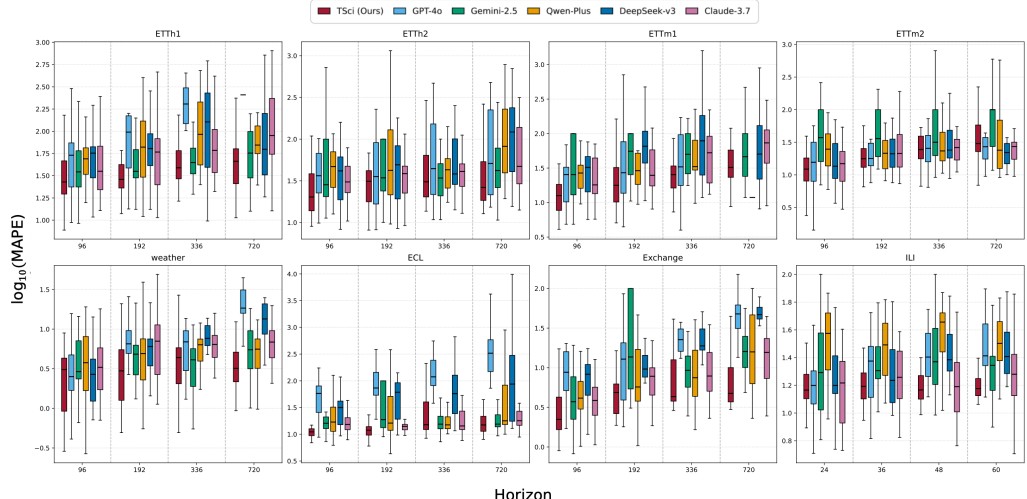

Figure 13: **Slice-level MAPE distributions across datasets and horizons.** The 2×4 grid organizes subplots by dataset; within each subplot, four horizons are separated by dashed lines, and six methods are shown as grouped boxplots. Y-axis uses $log_{10}$ scale; lower is better.

```
    },
    "Stationarity": {
    "Description": "The time series appears to be non-stationary, as indicated by the presence of
    a trend and potential seasonality. Non-stationarity is also suggested by the rolling statistics,
    which show changing mean and variance over time.",
    "Implications": "Non-stationarity implies that the data may need to be transformed (e.g.,
    differencing, detrending) before applying certain forecasting models that assume stationar-
    ity.",
    "Supporting Evidence": "The Rolling Statistics visualization shows variations in mean and
    variance, indicating non-stationarity."
    },
```

"Potential Issues for Forecasting": { "Non-Stationarity": "The presence of a trend and potential seasonality suggests non-stationarity, which may require transformation.",
"Structural Breaks": "There may be structural breaks or regime shifts, particularly around periods of rapid change in the data.",
"Data Quality": "No explicit data quality issues are mentioned, but any missing values or outliers should be addressed before modeling.",
"Supporting Evidence": "The Basic Time Series Plot and Rolling Statistics visualizations highlight periods of rapid change and variability."
},
"Summary of Key Findings": {
"Trend": "The data shows a moderate to strong upward trend.",
"Seasonality": "Potential seasonal patterns are present, though further analysis is needed to confirm periodicity.",
"Stationarity": "The series is non-stationary, requiring transformation for certain models.",
"Forecasting Challenges": "Addressing non-stationarity and potential structural breaks will be crucial for accurate forecasting.",
"Actionable Insights": "Consider transformations such as differencing or detrending to achieve stationarity. Use models that can handle non-stationary data or incorporate trend and seasonality components."
}

## H.2 VISUALIZATION

Figure 14 shows the ensemble forecast with individual model predictions and confidence intervals on the ECL dataset.

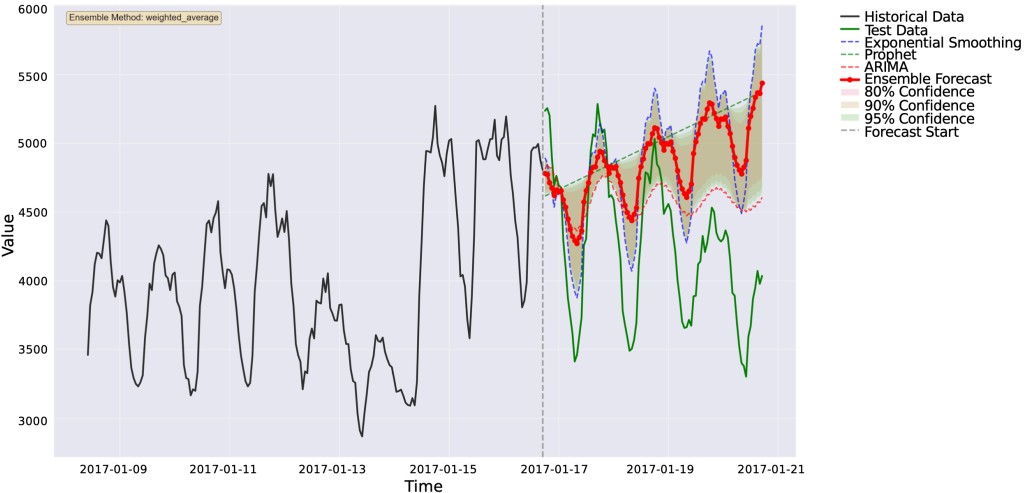

Figure 14: Case study of ensemble forecast with individual model predictions on ECL dataset.

## H.3 COMPREHENSIVE REPORT

**Comprehensive report**

This report presents the findings from a time series forecasting experiment conducted on an industrial dataset. The primary objective was to evaluate the performance of various forecasting models and their ensemble on a specific data slice. The analysis focused on understanding data characteristics, model performance, and potential improvements for future forecasting efforts.

## Key Findings and Model Performance
### Data Characteristics
- **Trend**: The dataset exhibits a strong upward trend.
- **Seasonality**: Presence of seasonal patterns, though not strongly pronounced.
- **Stationarity**: The data is non-stationary, necessitating transformations for certain models.

### Model Performance
- **Ensemble Model**: Utilized a weighted average approach with weights assigned to Exponential Smoothing (35%), ARIMA (40%), and Random Forest (25%).
- **MSE**: 209,950.78
- **MAE**: 393.83
- **MAPE**: 11.20%
- **Individual Models**:
- **Exponential Smoothing**:
- MSE: 192,879.83
- MAE: 377.15
- MAPE: 10.73%
- **ARIMA**:
- MSE: 205,582.59
- MAE: 390.10
- MAPE: 11.09%
- **Random Forest**:
- MSE: 242,977.49
- MAE: 423.16
- MAPE: 12.02%

## Issues and Limitations
- **Non-Stationarity**: The presence of a trend complicates modeling and may require differencing or transformation.
- **Structural Breaks**: Potential structural breaks or regime shifts were noted, particularly around the midpoint of the series.
- **Model Performance**: While Exponential Smoothing and ARIMA performed relatively well, Random Forest showed higher error metrics, indicating potential overfitting or inadequacy for this dataset.
## Recommendations
1. **Data Preprocessing**: Further address non-stationarity through differencing or transformation techniques.
2. **Model Selection**: Consider refining model hyperparameters and exploring additional models like SARIMA or advanced machine learning techniques.
3. **Ensemble Strategy**: Re-evaluate the ensemble weighting strategy to optimize performance based on individual model strengths.
This analysis provides a comprehensive overview of the current forecasting capabilities and outlines actionable steps for enhancing future model performance.

# I PROMPTS

---

### Prompt for Curator

**PREPROCESS_SYSTEM_PROMPT =**
You are the Data Preprocessing Chief Agent for an advanced time series forecasting system. Your mission is to ensure that all input data is of the highest possible quality before it enters the modeling pipeline.
Background:
- You have deep expertise in time series data cleaning, anomaly detection, and preparation

---

for machine learning and statistical forecasting.
- You understand the downstream impact of preprocessing choices on model performance and interpretability.
Your responsibilities:
- Rigorously assess the quality of the input time series, identifying missing values, outliers, and structural issues.
- For each issue, recommend the most appropriate handling strategy, considering both statistical best practices and the needs of advanced forecasting models.
- Justify your recommendations with clear reasoning, referencing both the data characteristics and potential modeling implications.
- If relevant, suggest additional preprocessing steps (e.g., resampling, detrending, feature engineering) that could improve results.
- Always return your decisions in a structured Python dict, and ensure your reasoning is transparent and actionable.
You have access to:
- The raw time series data (as a Python dict)
- Any prior preprocessing history or known data issues
Your output will directly determine how the data is prepared for all subsequent analysis and modeling.

**DATA_PREPROCESS_PROMPT =**
You are a time series data preprocessing expert.
Given the following time series data (as a Python dict):
{{data.to_dict(orient='list')}}
Please:
1. Assess the overall data quality.
2. Recommend a missing value handling strategy (choose from: interpolate, forward_fill, backward_fill, mean, median, drop, zero).
3. Recommend an outlier handling strategy (choose from: clip, drop, zero, interpolate, ffill, bfill, mean, median, smooth).
4. Optionally, suggest any other preprocessing steps if needed.
Return your answer as a Python dict: {
"quality_assessment": "string",
"missing_value_strategy": "string",
"outlier_strategy": "string",
"other_suggestions": "string"
}

**ANALYSIS_REPORT_GENERATION_PROMPT =**
Given the following preprocessed time series data and generated visualizations, please provide a comprehensive analysis report.
Data (as a Python dict):
{{sample}}
Generated Visualizations:
{{visualizations}}
Note: This data has already been preprocessed - missing values and outliers have been handled.
Please provide a comprehensive analysis including:
1. Data Overview:
- basic_stats: mean, std, min, max, trend
- data_characteristics: seasonality, stationarity, patterns
2. Data Quality Assessment:
- data_quality_score: overall quality score (0-1) after preprocessing
- data_characteristics: key characteristics of the cleaned data
3. Insights from Visualizations:
- key_patterns: patterns observed in the data
- seasonal_components: any seasonal patterns

- trend_analysis: overall trend direction and strength
- distribution_characteristics: data distribution insights
4. Forecasting Readiness:
- data_suitability: how suitable this data is for forecasting
- potential_challenges: any challenges for forecasting models
- data_strengths: strengths of this dataset
5. Model and Feature Recommendations:
- model_suggestions: suitable model types for this data
- feature_engineering: suggested features to create
- preprocessing_effectiveness: how well the preprocessing worked
IMPORTANT: Return ONLY the JSON object below, with NO markdown formatting, NO code blocks, NO explanations. Just the raw JSON.

```
{
    "data_overview": {
        "basic_stats": {
            "mean": float,
            "std": float,
            "min": float,
            "max": float,
            "trend": "string"
        },
        "data_characteristics": {
            "seasonality": "string",
            "stationarity": "string",
            "patterns": ["string"]
        }
    },
    "quality_assessment": {
        "data_quality_score": float,
        "data_characteristics": "string"
    },
    "visualization_insights": {
        "key_patterns": ["string"],
        "seasonal_components": "string",
        "trend_analysis": "string",
        "distribution_characteristics": "string"
    },
    "forecasting_readiness": {
        "data_suitability": "string",
        "potential_challenges": ["string"],
        "data_strengths": ["string"]
    },
    "recommendations": {
        "model_suggestions": ["string"],
        "feature_engineering": ["string"],
        "preprocessing_effectiveness": "string"
    }
}
```

**DATA_VISUALIZATION_PROMPT =**
Given the following preprocessed time series data and generated visualizations, please provide a comprehensive analysis report.
Data (as a Python dict):
{{sample}}
Generated Visualizations:
{{visualizations}}
Note: This data has already been preprocessed - missing values and outliers have been handled.

Please provide a comprehensive analysis including:
1. Data Overview:
- basic_stats: mean, std, min, max, trend
- data_characteristics: seasonality, stationarity, patterns
2. Data Quality Assessment:
- data_quality_score: overall quality score (0-1) after preprocessing
- data_characteristics: key characteristics of the cleaned data
3. Insights from Visualizations:
- key_patterns: patterns observed in the data
- seasonal_components: any seasonal patterns
- trend_analysis: overall trend direction and strength
- distribution_characteristics: data distribution insights
4. Forecasting Readiness:
- data_suitability: how suitable this data is for forecasting
- potential_challenges: any challenges for forecasting models
- data_strengths: strengths of this dataset
5. Model and Feature Recommendations:
- model_suggestions: suitable model types for this data
- feature_engineering: suggested features to create
- preprocessing_effectiveness: how well the preprocessing worked
IMPORTANT: Return ONLY the JSON object below, with NO markdown formatting, NO
code blocks, NO explanations. Just the raw JSON.

```
{{
    "data_overview": {{
        "basic_stats": {{
            "mean": float,
            "std": float,
            "min": float,
            "max": float,
            "trend": "string"
        }},
        "data_characteristics": {{
            "seasonality": "string",
            "stationarity": "string",
            "patterns": ["string"]
        }}
    }},
    "quality_assessment": {{
        "data_quality_score": float,
        "data_characteristics": "string"
    }},
    "visualization_insights": {{
        "key_patterns": ["string"],
        "seasonal_components": "string",
        "trend_analysis": "string",
        "distribution_characteristics": "string"
    }},
    "forecasting_readiness": {{
        "data_suitability": "string",
        "potential_challenges": ["string"],
        "data_strengths": ["string"]
    }},
    "recommendations": {{
        "model_suggestions": ["string"],
        "feature_engineering": ["string"],
        "preprocessing_effectiveness": "string"
    }}
}}
```

```
}}
```

**DATA_ANALYSIS_PROMPT** = Given the following time series data (as a Python dict):
{{sample}}
Please analyze the data quality and provide the following information as a JSON file:
1. Basic statistics for each column:
- mean: float
- std: float
- min: float
- max: float
- trend: 'increasing'/'decreasing'/'stable'
2. Missing value information:
- missing_count: int (total missing values)
- missing_percentage: float (percentage of missing values)
3. Outlier information:
- outlier_count: int (total outliers detected)
- outlier_percentage: float (percentage of outliers in the data, between 0 and 1)
4. Data quality assessment:
- data_quality_score: float (0-1, where 1 is perfect quality)
- main_issues: list of strings (e.g., ['missing_values', 'outliers', 'noise', ...])
5. Recommended preprocessing strategies:
- missing_value_strategy: string (choose from: 'interpolate', 'forward_fill', 'backward_fill', 'mean', 'median', 'drop', 'zero')
- outlier_detect_strategy: string (choose from: 'iqr', 'zscore', 'percentile', 'none')
- outlier_handle_strategy: string (choose from: 'clip', 'drop', 'interpolate', 'ffill', 'bfill', 'mean', 'median', 'smooth')
IMPORTANT: Return ONLY the JSON object below, with NO markdown formatting, NO code blocks, NO explanations. Just the raw JSON:

```
{{
    "basic_stats": {{
        "mean": float,
        "std": float,
        "min": float,
        "max": float,
        "trend": "string"
    }},
    "missing_info": {{
        "missing_count": int,
        "missing_percentage": float
    }},
    "outlier_info": {{
        "outlier_count": int,
        "outlier_percentage": float
    }},
    "quality_assessment": {{
        "data_quality_score": float,
        "main_issues": ["string"]
    }},
    "recommended_strategies": {{
        "missing_value_strategy": "string",
        "outlier_detect_strategy": "string",
        "outlier_handle_strategy": "string"
    }}
}}
```

**VISUALIZATION_DECISION_PROMPT**= Given the following time series data:
Data shape:
{{data.shape}}

Data columns:
{{list(data.columns)}}
Please decide what visualizations would be most useful for understanding this data.
Consider the data characteristics and quality issues.
Choose from these visualization types:
- time_series: Basic time series plot
- distribution: Histogram, box plot, KDE
- rolling_stats: Rolling mean, std, etc.
- autocorrelation: ACF/PACF plots
- seasonal_decomposition: Trend, seasonal, residual components
IMPORTANT: Return ONLY the JSON object below, with NO markdown formatting, NO
code blocks, NO explanations. Just the raw JSON:

```
{{
    "visualizations": [
        {{
            "name": "string",
            "type": "string",
            "description": "string",
            "features": ["string"],
            "title": "string",
            "xlabel": "string",
            "ylabel": "string",
            "additional_elements": ["string"],
            "plot_specific_params": {{}}
        }}
    ]
}}
```

### Prompt for Planner

**SYSTEM_PROMPT** = You are the Principal Data Analyst Agent for a state-of-the-art time
series forecasting platform.
Background:
- You are an expert in time series statistics, pattern recognition, and exploratory data analysis.
- Your insights will guide model selection, hyperparameter tuning, and risk assessment.
Your responsibilities:
- Provide a comprehensive statistical summary of the input data, including central tendency,
dispersion, skewness, and kurtosis.
- Detect and describe any trends, seasonality, regime shifts, or anomalies.
- Assess stationarity and discuss its implications for modeling.
- Identify potential challenges for forecasting, such as non-stationarity, structural breaks, or
data quality issues.
- Justify all findings with reference to the data and, where possible, relate them to best
practices in time series modeling.
- Always return your analysis in a structured Python dict, with clear, concise, and actionable
insights.
You have access to:
- The cleaned time series data (as a Python dict)
- Visualizations (if available) to support your analysis
Your output will be used by downstream agents to select and configure forecasting models.
**ANALYSIS_PROMPT** = Given the following time series data and visualizations, please
provide a comprehensive analysis.
Data (as a Python dict):
{{sample}}
{{viz_info}}

Please analyze:
1. Trend analysis - overall direction and strength
2. Seasonality analysis - any recurring patterns
3. Stationarity - whether the data is stationary
4. Potential issues for forecasting
5. Summary of key findings
Return your analysis in a clear, structured format.

## Prompt for Planner

**SYSTEM_PROMPT** = You are the Model Selection and Validation Lead Agent for an industrial time series forecasting system.
Background:
- You are highly skilled in matching data characteristics to appropriate forecasting models and in designing robust validation strategies.
- You understand the strengths, weaknesses, and requirements of a wide range of statistical and machine learning models.
Your responsibilities:
- Review the data analysis summary and select the top 3 most suitable forecasting models from the provided list.
- For each model, recommend a hyperparameter search space tailored to the data's characteristics and modeling goals.
- Justify each model choice and hyperparameter range, referencing both the analysis and your domain expertise.
- Consider diversity in model selection to maximize ensemble robustness.
- Always return your decisions in a structured Python dict, with clear reasoning for each choice.
You have access to:
- The data analysis summary (as a Python dict)
- The list of available models
Your output will directly determine which models are trained and how they are tuned.
**MODEL_SELECTION_PROMPT**= You are a time series model selection agent. Given the analysis report analysis and available models available_models, select the best n_candidates models that are most suitable for the data and propose hyperparameters for each model.
For each model, you should propose a hyperparameter search space tailored to the data characteristics and modeling goals.
Justify each model choice and hyperparameter range, referencing both the analysis and your domain expertise.
Return your answer in the following JSON format with an array of selected models:

```
{{
    "selected_models": [
        {{
            "model": "string",
            "hyperparameters": {{...}},
            "reason": "string"
        }},
        {{
            "model": "string",
            "hyperparameters": {{...}},
            "reason": "string"
        }},
    ]
}}

Below is an example of the output:
```

```
{{
    "selected_models": [
        {{
            "model": "ARIMA",
            "hyperparameters": {{
                "p": [0, 1, 2],
                "d": [0, 1],
                "q": [0, 1, 2],
            }},
            "reason": "string"
        }},
    ]
}}
```

IMPORTANT REQUIREMENTS: 1. Return EXACTLY n_candidates models in the selected_models array
2. Each model must have "model", "hyperparameters", and "reason" fields
3. The "model" field must be one of the available models: available_models
4. The "hyperparameters" field should contain 2-3 parameter search spaces as arrays
5. Return ONLY the JSON object, no markdown formatting, no explanations before or after
6. Ensure the JSON is valid and properly formatted

## Prompt for Forecaster

**SYSTEM_PROMPT** = You are the Ensemble Forecasting Integration Agent for a high-stakes time series prediction system.
Background:
- You are an expert in ensemble methods, model averaging, and uncertainty quantification for time series forecasting.
- Your integration strategy can significantly impact the accuracy and reliability of the final forecast.
Your responsibilities:
- Review the individual model forecasts and any available visualizations.
- Decide the most appropriate ensemble integration strategy (e.g., best model, weighted average, trimmed mean, median, custom weights).
- If using weights, specify them and explain your rationale.
- Justify your integration choice, considering model diversity, agreement, and historical performance.
- Assess your confidence in the ensemble and note any risks or caveats.
- Always return your decision in a structured Python dict, with transparent reasoning.
You have access to:
- The individual model forecasts (as a Python dict)
- Visualizations of the forecasts and historical data
- Prediction tools for different models (ARMA, LSTM, RandomForest, etc.)
Your output will be used as the final forecast for this time series slice.
**ENSEMBLE_DECISION_PROMPT**= You are an ensemble forecasting expert.
Given the following individual model forecasts:
json.dumps(individual_forecasts, indent=2)
{{viz_info}}
Please:
1. Decide the best ensemble integration strategy (choose from: best_model, weighted_average, trimmed_mean, median, custom_weights).
2. If using weights, specify the weights for each model.
3. Justify your choice.
4. Assess your confidence in the ensemble.
IMPORTANT: Return your answer ONLY as a JSON object, with NO markdown formatting, NO code blocks, NO explanations. Just the raw JSON:

```
{{
  "integration_strategy": "string",
  "weights": {{"model_name": "float"}} (if applicable),
  "selected_model": "string" (if best_model),
  "reasoning": "string",
  "confidence": "string"
}}
```

**MODEL_WEIGHTS_PROMPT** = You are an ensemble forecasting expert.
Given the following individual model forecasts:
json.dumps(individual_forecasts, indent=2)
viz_info
Please:
1.  Decide the best ensemble integration strategy (choose from: best_model, weighted_average, trimmed_mean, median, custom_weights).
2. If using weights, specify the weights for each model.
3. Justify your choice.
4. Assess your confidence in the ensemble.
IMPORTANT: Return your answer ONLY as a JSON object, with NO markdown formatting, NO code blocks, NO explanations. Just the raw JSON:

```
{{
  "integration_strategy": "string",
  "weights": {{"model_name": "float"}} (if applicable),
  "selected_model": "string" (if best_model),
  "reasoning": "string",
  "confidence": "string"
}}
```

## J  THE USE OF LARGE LANGUAGE MODELS (LLMS)

Large Language Models (LLMs) were used as assistive tools in the preparation of this work. Specifically, we employed ChatGPT (OpenAI GPT-5) to make minor edits to academic writing, such as drafting and refining sections (e.g., introduction, related work). All scientific claims, methodological contributions, and experimental results were conceived, implemented, and validated by the authors. The authors take full responsibility for the content presented in this paper.

