# OpenReview forum: "A Stitch in Time Forecasts Nine: Towards End-to-End Agentic Time Series Forecasting"
_ICLR.cc/2026/Conference — ICLR 2026 Conference Withdrawn Submission_

### Official Review · Reviewer_FQ99 · 2025-10-26

**Soundness:** 2
**Presentation:** 2
**Contribution:** 1
**Rating:** 2
**Confidence:** 5

**Summary:**

Generating time series forecasting has many practical applications in across the field. This paper claims that they propose a first so called multi-agent system (4 agents) and an agentic TS analysis. The paper has presented an in-depth experimental analysis and the comparative study, however, the paper miss many important existing literature works which which the proposed method may need to compare and scientifically evaluate.

**Strengths:**

- The problem of building a TSScientist is a need from end user point of view who are mainly business persona.
- The conducting experiments on various LLM demonstrate the benefits of multi-agent system for time series work.

**Weaknesses:**

1. The major concerns I have is paper failed to capture the existing literature works :
- DSAgent
- MLEBench
- R&D Agent
- MLEStar
- .. any more
A douzen of paper has proposed a scientific agent discovery on time series data, they all can be used one a way to build a time series forecasting model and also adopted a multi-agent sytem approach, doing data reporting and etc.

2. CodeAsAgent such as MetaGPT can actually write a nice ML/DS code and it can be a strong baseline.

3. Paper does keep mention of Reasoning, but it is not been discussed in paper at great detail. such as is this a Chain of Thoguth, and some thing esle?

**Questions:**

I have provided three points above incase author like to justify the ommission of these appropriate related work.

---

### Official Review · Reviewer_utb8 · 2025-10-28

**Soundness:** 1
**Presentation:** 2
**Contribution:** 2
**Rating:** 2
**Confidence:** 5

**Summary:**

This paper proposes a multi-LLM-agent framework for time series forecasting that coordinates data preprocessing, model selection, and parameter tuning. The direction is meaningful, though the current experiments rely on established, clean, context-free benchmark datasets and do not demonstrate the practical benefits claimed for real-world forecasting scenarios.

**Strengths:**

- Meaningful direction in time series domains that emphasizes real-world forecasting workflows and multi-agent systems.

- Promising idea of using an agent for plot-informed multimodal diagnostics.

**Weaknesses:**

- The proposed framework is conceptually comprehensive, but the experiments remain limited to traditional and well-structured datasets. These settings fail to validate whether the system can truly address real-world forecasting challenges (data quality issues, regime shifts, missing context). Current results do not sufficiently support the practical claims.

- Gap between the narrative and the actual engineering difficulty. I agree with the authors’ statement that: “The dominant cost in forecasting is not model fitting, but building reliable data processing and evaluation pipelines.” However, the paper does not demonstrate how the proposed agent improves the actual pipeline effort. The benefit remains largely theoretical.

- Multimodal diagnostics need more substantial evidence. ''Plot-informed multimodal diagnostics'' is an appealing component, yet the paper does not show clear examples or tangible improvements enabled by this capability.

- Limited robustness to distribution shifts: Real data contains shocks, sudden trend reversals, and structural breaks. It is unclear how the agent responds to abrupt changes or anomalies that do not align with historical patterns.

- Performance ceiling constrained by model pool. The main performance leverage is model selection, but the upper bound still depends on the candidates’ capability. How does the proposed approach exceed the best candidate model’s limit? Can the authors leverage LLM reasoning beyond simple selection/tuning?

- Time cost is not discussed. The framework introduces additional steps (dialog, diagnostics, reasoning), but how much overhead does it add? Is it deployable in operational forecasting?

- Interpretability is not equal to actionable reporting. The explanations are readable, but do they drive better decision-making, or replace domain expertise? More evaluation on interpretability utility is needed.

**Questions:**

1. Current experiments rely solely on conventional benchmarks. Can the authors evaluate on newer and more realistic forecasting tasks, such as Context is Key (CiK), to better validate real-world claims?
   - A. R. Williams et al., *Context is Key: A Benchmark for Forecasting with Essential Textual Information*, arXiv:2410.18959 (ICML 2025).

2. How do multimodal diagnostics and agent reasoning concretely improve decision-making, beyond generating post-hoc interpretations?

3. What is the computational overhead of the agent process, and is the approach deployable for real-time or operational forecasting workloads?

---

### Official Review · Reviewer_BmyG · 2025-10-29

**Soundness:** 1
**Presentation:** 2
**Contribution:** 1
**Rating:** 0
**Confidence:** 5

**Summary:**

This paper presents TimeSeriesScientist (TSci), an end-to-end, agentic framework that automates the univariate time series forecasting workflow. TSci contains four agents: Curator used to diagnose the data and preprocess the data accordingly; Planner proposes the optimal models and their hyperparameter configuration; Forecasting constructs an ensemble based on the optimized models; Reporter provides an analysis of the forecasting results. Results show that TSci could outperform statistical and LLM-driven baselines on eight long-term forecasting datasets.

**Strengths:**

Overall, this paper is well-written and easy to follow. The idea of proposing an end-to-end forecasting workflow that combines outlier detection, forecasting, and analysis is also interesting. The LLM interface and reporter could provide a better overview for the users.

**Weaknesses:**

TLDR: Yet another agentic paper without any substantial novelty.

My biggest concern is the role that an LLM plays in this agentic system. Overall, except for acting as a user-framework interface, the only part that requires the knowledge of an LLM is to recommend the missing value and outlier handling strategies, select $n_p$ models from the model pool, and proposing the ensemble strategy, as the other parts only run with fixed rules that do not need the knowledge from LLM. Even the model selection parts can be learned through meta-learning. It is questionable if the LLM is really required in this scenario. And therefore, the main contribution of this paper is still unclear to me.

Since the LLM knowledge is only used in the model selection part, an ablation study is missing, i.e., if the LLM will always select the same outlier handling strategies, the same set of models, and the same ensemble strategy. What would be the impact of changing these strategies to another one or sampling them randomly?

Although the authors claim that the TSci could automate the entire workflow for time-series forecasting, it still requires many hyperparameters. For instance, the $\alpha$ values in Eq. 5, 6, the $\beta, \tau, \lambda, \omega_{min}$ in Eq. 21-23, and the window size $W_t$. All these hyperparameters might also be critical to the decision-making process, but their impacts are not properly evaluated. (In addition, it is quite well known that LLMs have problems in setting hyperparameters well beyond established benchmarks covered in their training data.)

 As far as I know, the datasets evaluated in this paper (ECL, weather, Exchange, ILI, ETTs) do not contain missing values. It is also interesting to see how the approach performs on datasets with missing values if TSci already implements missing value handling strategies.

The dataset only contains the multi-variant long-term forecasting datasets. However, since this paper aims at univariate forecasting tasks, it should also consider the univariate forecasting datasets such as M4 datasets [1]

There are many ways to apply LLM to forecasting tasks [2,3]. However, it is unclear how the baseline LLMs are applied to forecasting tasks in this paper.

Many important baselines, such as PatchTST [4] and other foundational forecasting models [5,6], are not evaluated.

The model pool mainly involves the statistical and traditional ML-based models and misses the recently proposed deep learning forecasting models, such as DeepAR [7], N-BEATS [8], and PatchTST[4].

Last but not least, it is unclear whether the approach is doing a simple information retrieval since the datasets are already known to the LLM at training time. I strongly doubt that this is a valid evaluation in the first place.

[1] Makridakis et al, The M4 Competition: 100,000 time series and 61 forecasting methods
[2] Gruver et al. Large Language Models Are Zero-Shot Time Series Forecasters
[3] Jin et al.Time-LLM: Time Series Forecasting by Reprogramming Large Language Models
[4] Nie et al, A Time Series is Worth 64 Words: Long-term Forecasting with Transformers
[5] Das et al, A decoder-only foundation model for time-series forecasting
[6] Ansari et al, Chronos: Learning the Language of Time Series
[7] Salinas et al, DeepAR: Probabilistic Forecasting with Autoregressive Recurrent Networks
[8] Oreshkin et al, N-BEATS: Neural basis expansion analysis for interpretable time series forecasting

**Questions:**

I have several questions regarding the hyperparameters applied in this paper

* Does the LLM only propose the candidate pool, or does it also involve the hyperparameter configuration sampling process? If so, does the LLM sample configurations sequentially or multiple configurations at once? If no, are the configurations sampled randomly or sampled via some optimization process (e.g., Bayesian optimization)?
* Since the model pool only contains 21 models, what is the size of the candidate pool (line 247), and how many samples are sampled from each model family?
* How to check if one report wins another one for each rubrics reported in Table 2 (and appendix F), is it judged by another LLM or by humans?

---

### Official Review · Reviewer_twNY · 2025-11-01

**Soundness:** 2
**Presentation:** 3
**Contribution:** 1
**Rating:** 4
**Confidence:** 4

**Summary:**

The paper proposes TimeSeriesScientist (TSci), a four-agent framework (Curator, Planner, Forecaster, Reporter) that automates an end-to-end pipeline for univariate time-series forecasting.

1. Curator performs leakage-aware diagnostics and preprocessing;
2. Planner selects and tunes models from a fixed library
3. Forecaster ensembles tuned candidates;
4. Reporter compiles forecasts, metrics, visualizations, and rationales.

Experiments across eight public benchmarks (ETTh1/2, ETTm1/2, Weather, Electricity, Exchange, ILI) compare TSci primarily to zero-shot LLM forecasters (GPT-4o, Gemini-2.5 Flash, Qwen-Plus, DeepSeek-v3, Claude-3.7), with supplementary plots vs a few statistical baselines. TSci uses GPT-4o as the default backbone and, “due to a limited budget,” evaluates on 25 slices per dataset (T=512, horizons $H \in {96,192,336,720}$), reporting slice-averaged MAE/MAPE. The paper also evaluates the generated reports via five rubrics (AS, MJ, IC, AQ, SC) using an LLM judge for pairwise comparisons, claiming higher win rates than LLM baselines.

**Strengths:**

* The paper is well-structured (framework overview, modules, ablation, case study) and generally clear. Interpretable Reporter that surfaces metrics, confidence intervals, and rationales. Figures are informative, the agent roles and pipeline are easy to follow. Minor issues: i) Strong claims (“dominant cost… not model fitting,” “white-box system”) would benefit from supporting references/quantification. ii) Typo: Figure 3 central graphic reads “Externa Tools” (should be “External Tools”). (Figure 3 is the Curator workflow.)
* The paper documents leakage-safe preprocessing (rolling stats, causal windows, clear train/val/test discipline for ensembling), ablations across modules (show sizable degradations when removing preprocessing/analysis/optimization), and a coherent multi-agent protocol.
* Clear modular design (Curator/Planner/Forecaster/Reporter) that mirrors human workflows. Good leakage awareness in preprocessing.
* Coherent narrative + ablations: removing any module hurts performance; this supports the architectural decomposition.
* Usability angle: Reporter compiles visualizations, metrics, and rationales into comprehensive outputs, which could aid auditability in practice.
* Breadth of datasets: Eight benchmarks across several domains (electricity, environment, macro, health).

**Weaknesses:**

Soundness
---------------
The system design is reasonable and many leakage-avoidance choices are documented. However, central empirical claims aren’t yet sufficiently supported for ICLR. Key gaps:
* Baselines & fairness: Most headline tables compare to zero-shot LLMs, not to strong forecasting models or competitive AutoML toolkits. The statistical comparisons are limited and not the focus of the main results. This makes the superiority claim over the SOTA forecasting landscape insufficiently justified.
* Backbone dependence: TSci relies on GPT-4o as the sole reasoning LLM; there is no sensitivity analysis replacing it with multiple models (e.g., Claude, Qwen, DeepSeek) to test robustness or cost–performance trade-offs.
* Evaluation protocol and budget: The authors process only 25 slices per dataset “due to a limited budget.” It’s unclear which budget (compute vs monetary tokens), how token/compute usage scales with series length, and whether LLM baselines and statistical baselines were evaluated under the same slice protocol and token constraints. The paper states consistent protocols, but details of prompting/cost parity are missing.
* Report-quality evaluation: The report rubrics (AS, MJ, IC, AQ, SC) are defined, but the judge model and prompts used to score reports are not fully specified in the main paper, therefore reproducibility is unclear. (Appendix not includes the judging prompt.)
* Interpretability claim: The introduction’s statement that TSci yields a “white-box” system is overstated, since decisions depend on a closed-weight LLM (GPT-4o). natural-language rationales does not imply mechanistic transparency.

Contribution
----------------
While an agentic, end-to-end pipeline is timely and practically useful, the novelty and significance are limited relative to ICLR standards:
* The framework bundles existing components (diagnostics, model selection from a standard library, ensembling, LLM rationales) rather than introducing a new learning principle or representation.
* The scope is univariate forecasting; no multivariate or irregular multimodal results, which would be a more convincing step forward for ICLR.
* Without strong SOTA baselines beyond zero-shot LLMs, it’s hard to assess contribution to the forecasting literature proper.

General
-----------
* Baseline scope and fairness. Headline comparisons center on zero-shot LLMs; strong forecasting baselines are missing from main tables, making the superiority claim incomplete.
* Reasoning LLM ablation missing. Results are reported only with GPT-4o; replacing it with multiple backbones (Claude/Qwen/DeepSeek) would test robustness and cost.
* Budget and slicing: The 25-slice protocol is under-explained (which budget? why 25? token accounting?), and it’s unclear whether all methods, including statistical ones, were subject to identical resource and token constraints.
* Interpretability overclaim: Calling the system “white-box” is misleading when crucial decisions are produced by a proprietary LLM. The paper demonstrates explanations, not introspectable mechanisms.
* Report scoring transparency: The judge LLM and prompts for REPORT EVALUATION RUBRICS aren’t specified; reproducibility and bias controls are unclear.
* Scope limited to univariate: No multivariate/irregular series, though those are common and more impactful. This limits generality.
* Claim substantiation: The statement that the dominant cost is not model fitting lacks quantitative evidence or citations specific to cost breakdowns in practice.
* Originality and significance caveat: multi-agent orchestration for TS forecasting is interesting but not unique; given current activity around agentic pipelines, incremental novelty + limited baselines reduce the perceived significance.

I’m positive on the idea and usability, but current experiments and baselines are not sufficient for ICLR-level claims. A strong rebuttal with expanded baselines, backbone sensitivity, clearer evaluation protocol, and toned-down interpretability claims could move this to accept. (If updates are not feasible, my fallback would be 2: reject, not good enough for this venue.)

**Questions:**

* Baselines: Will you add tables vs state-of-the-art forecasting models (e.g., PatchTST, N-HiTS, TimesFM/Chronos; strong AutoML like AutoGluon-TS) with the exact prompts/hyperparameters used for each baseline and leakage controls?
* Backbone sensitivity: Can you swap the reasoning LLM (Claude-3.7, Qwen-Plus, DeepSeek-v3) throughout TSci and report accuracy and token/latency cost, to show that gains aren’t GPT-4o-specific?
* Slicing protocol: What budget constrained experiments (monetary tokens, compute time)? Why 25 slices, and how does accuracy change with more slices or full datasets? Provide token-level cost accounting per method and confirm identical slice usage across baselines.
* Prompt details for baselines: For each zero-shot LLM baseline, provide the exact prompts, temperature, stopping criteria, and token caps. Otherwise the comparisons risk being non-reproducible.
* Report evaluation: Which LLM judge and prompt were used for the five rubrics (AS, MJ, IC, AQ, SC)? Any calibration (e.g., swap judge models, judge self-preference checks, position bias countermeasures)?
* Single- vs multi-agent ablation: Please compare TSci to a single-agent variant with identical tools to justify the multi-agent design beyond modularity.
* Data efficiency: Show performance vs training-window length T and vs number of slices, and add experiments on few-series regimes.
* Multivariate extension: Can you report at least a pilot on multivariate datasets (even if scaled down) to evidence broader applicability?
* Cost dominance claim: Provide empirical evidence (time/money logs) supporting the assertion that pipeline building, not model fitting, is the dominant cost in practice.
* Reporter trustworthiness: The Reporter seems promising for explainability; can you quantify user trust/utility (e.g., human evals) and analyze failure cases where rationales conflict with chosen models?

**Details Of Ethics Concerns:**

No ethics concerns.

---

### Note · Authors · 2025-11-12

I have read and agree with the venue's withdrawal policy on behalf of myself and my co-authors.